# Genome-Wide Characterization of the *ANN* Gene Family in *Corydalis saxicola* Bunting and the Role of CsANN1 in Dehydrocavidine Biosynthesis

**DOI:** 10.3390/plants14131974

**Published:** 2025-06-27

**Authors:** Han Liu, Jing Wang, Zhaodi Wen, Mei Qin, Ying Lu, Lirong Huang, Xialian Ou, Liang Kang, Cui Li, Ming Lei, Zhanjiang Zhang

**Affiliations:** 1Guangxi Key Laboratory of Medicinal Resources Protection and Genetic Improvement, Guangxi Botanical Garden of Medicinal Plants, Nanning 530023, China; gmhg_lh@163.com (H.L.); 18565756016@139.com (J.W.); w2267318207@163.com (Z.W.); qinmei20210630@163.com (M.Q.); 15807898547@163.com (Y.L.); huanglirong2066@126.com (L.H.); 18290045835@163.com (X.O.); 17736623451@163.com (L.K.); licuicui941@163.com (C.L.); 2National Center for Traditional Chinese Medicine Inheritance and Innovation, Guangxi Botanical Garden of Medicinal Plants, Nanning 530023, China; 3Guangxi Engineering Research Center of Traditional Chinese Medicine Resource Intelligent Creation, Guangxi Botanical Garden of Medicinal Plants, Nanning 530023, China; 4School of Pharmacy, Guangxi Medical University, Nanning 530021, China; 5Guangxi Key Laboratory for High-Quality Formation and Utilization of Dao-di Herbs, Guangxi Botanical Garden of Medicinal Plants, Nanning 530023, China

**Keywords:** annexin, *Corydalis saxicola* Bunting, calcium, benzylisoquinoline alkaloid

## Abstract

Annexins (ANNs) are a family of calcium (Ca^2+^)-dependent and phospholipid-binding proteins, which are implicated in the regulation of plant growth and development as well as protection from biotic and abiotic stresses. *Corydalis saxicola* Bunting, an endangered benzylisoquinoline alkaloid (BIA)-rich herbaceous plant, widely used in traditional Chinese medicine, is endemic to the calciphilic karst region of China. However, whether and how ANNs are involved in the biosynthesis pathway of BIAs and/or help *C. saxicola* plants cope with abiotic properties, such as calcareous soils, are largely unknown. Here, nine *CsANN* genes were identified from *C. saxicola*, and they were divided into three subfamilies, namely subfamilies I, II, and IV, based on the phylogenetic tree. The CsANNs clustered into the same clade, sharing similar gene structures and conserved motifs. The nine *CsANN* genes were located on five chromosomes, and their expansions were mainly attributed to tandem and whole-genome duplications. The *CsANN* transcripts displayed organ-specific and Ca^2+^-responsive expression patterns across various tissues. In addition, transient overexpression assays showed that CsANN1 could positively regulate the accumulation of BIA compounds in *C. saxicola* leaves, probably by directly interacting with key BIA-biosynthetic-pathway enzymes or by interacting with BIA-biosynthetic regulatory factors, such as MYBs. This study sheds light on the profiles and functions of the *CsANN* gene family and paves the way for unraveling the molecular mechanism of BIA accumulation, which is regulated by Ca^2+^ through CsANNs.

## 1. Introduction

*Corydalis saxicola* Bunting is an endangered herbaceous plant that belongs to the genus *Corydalis* within the Papaveraceae family [1]. In traditional Chinese medicine, *C. saxicola*, locally known as Yanhuanglian, was widely used to treat liver diseases, such as hepatitis [1], cirrhosis [2], liver cancer [3], as well as acute conjunctivitis and acute abdominal pain [3]. These pharmacological activities of Yanhuanglian are primarily attributed to the presence of benzylisoquinoline alkaloid (BIA) compounds, with dehydrocavidine (DHCA) identified as one of the key active ingredients [4,5]. As a species endemic to the karst region in China, *C. saxicola* lives exclusively in and around rock crevices and demonstrates good adaptability to the arid, infertile, and calcium-rich soils, which are the typical karst environments [6]. However, both its narrow, stringent karst-adapted habitat and the imbalance of large market demand render it an endangered species [6,7,8]. Therefore, understanding the adaptation mechanism of *C. saxicola* in calcium-rich karst habitats, as well as the regulatory molecular mechanism of the key active BIAs, such as DHCA, in *C. saxicola*, are considered as two of the most efficient and promising approaches for the conservation and utilization of this endangered medicinal plant.

Annexins (ANNs) are evolutionarily conserved, water-soluble proteins [9,10,11]. The plant ANNs, which can bind to or insert into cell membranes and regulate the homeostasis of free Ca^2+^ in the cytoplasm, are the members of voltage-gated Ca^2+^ channels [12,13]. Plant ANNs are characterized by a shorter N-terminal region, which comprises the main different amino acids and it is commonly the site for secondary modifications. In addition, there is a C-terminal region containing four conservative ANN repeats, among which repeats I and IV harbor Ca^2+^-binding sites [14,15,16,17]. Recently, the ANN families have been identified in many plant species, including *Carica papaya* [18], *Glycine max* [19], *Oryza sativa* [20], *Raphanus sativus* [21], *Vitis vinifera* [18], and *Zea mays* [22]. These ANNs are ubiquitously distributed across various plant tissues, including embryos [23], seedlings [17], roots and tubers [24,25,26], stems, hypocotyls and cotyledons, leaves, inflorescences, fruits, vascular systems, and phloem sap [27,28]. This demonstrates their wide involvements in regulating of biochemical and cellular processes, plant growth and development, and response to biotic and abiotic stresses [9,10,11,14,29,30,31].

Plants in karst areas generally have a tolerance to drought and they have calciphilic characteristics [32,33]. Our previous studies found that the cultivated *C. saxicola* plants had the ability to endure treatments with high concentrations of exogenous CaCl_2_ solutions for up to almost one month, and the levels of DHCA in the roots of *C. saxicola* were significantly enhanced after the treatments [6]. Nevertheless, the mechanism by which extracellular Ca^2+^ signaling was recognized and converted into intracellular ones to regulate the biosynthesis of DHCA in *C. saxicola,* is largely unknown. Similarly, the adaptation mechanism of wild *C. saxicola* to high calcium stress in karst areas also needs to be unraveled. As mentioned above, plant ANNs are members of voltage-gated Ca^2+^ channels and are involved in Ca^2+^ uptake and transport. Therefore, it was speculated that ANNs in *C. saxicola* (CsANNs) might also be important stress and BIA-biosynthesis regulators in *C. saxicola.*

To date, how plant ANNs, including CsANNs, are molecularly involved in the response to stress, as well as the regulation of secondary metabolite biosynthesis, is largely unknown. In this study, a global survey of the nine *ANN* genes in the *C. saxicola* genome was conducted. The physicochemical properties, conserved motifs, gene structures, and cis-elements in the promoter regions of *CsANN*s were identified. In addition, the homologous *ANN* genes from different species were determined by using collinearity analysis. Furthermore, the trascriptomic analysis and quantitative real-time PCR (qRT-PCR) results demonstrated that the *CsANN* transcripts displayed organ-specific and Ca^2+^-responsive expression patterns across various tissues. The transient overexpression assays indicated that CsANN1 could induce the accumulation of DHCA in the leaves of *C. saxicola*. The present study might provide substantial useful information for the understanding of CsANNs as the key effectors during Ca^2+^ uptake and transport, and the role played in the biosynthesis of DHCA in *C. saxicola*.

## 2. Results

### 2.1. Identification and Physicochemical Analysis of CsANNs

A total of nine candidate *ANN*s were identified in the *C. saxicola* genome. All these CsANNs contained conserved ANN domains, indicating that these peptides were putative ANNs. Subsequently, all nine *CsANN* genes were cloned by using PCR and Sanger sequencing verified the products. Consequently, these *CsANN*s were named *CsANN1-9* according to their chromosomal localization (Appendix A).

Physicochemical property analysis showed that the CsANN proteins contained 316 to 341 amino acids, with molecular weights ranging from 35.48 kDa to 38.67 kDa (Appendix A). The hydrophilicities of the nine CsANN proteins were all below 0, indicating their hydrophilic nature. The isoelectric points of CsANN1, 5, 6, 7, and 9 were all below 7, indicating that they were acidic hydrophilic proteins. The others were basic hydrophilic proteins. The protein instability indices of CsANN1, 3, 4, 5, and 7 were below 40, and these are stable proteins; the others were unstable proteins. The subcellular localization predictions for CsANNs, as determined by different software tools, exhibited variability (Figure 1A). CsANN1, 3, 5, and 7, 2, 6, and 8, 3, 4, 9 were predicted to be localized in the cytosol, chloroplasts, peroxisomes, extracellular matrix, and cytoskeleton, respectively (Figure 1A).

### 2.2. Sequences Alignment and Phylogenetic Analysis of ANNs

The amino acid sequence alignment results indicated that CsANN proteins contained four conserved ANN repeats, each of which consisted of approximately 70 amino acids and they were similar to that of AtANN1 (Figure 1B). In particular, the specific Ca^2+^-binding sites (G/KXGT-38-D/E), which could functionally bind to phospholipid membranes, were characterized in the repeats I of CsANN1, 5, 7, and 9 (Figure 1B). In addition, a conserved peroxidase residue (ILAHR) was identified in CsANN1, 5, and 9, indicating that they were likely to have peroxidase activity. Furthermore, the interaction region indicator (IRI) site, which could bind to filamentous actin (F-actin), existed in the repeat III region of CsANN1 and 9, as well as the DXXG site, which could bind to GTP, and it was found in the repeat IV of CsANN1, 5, and 9 (Figure 1B).

To elucidate the evolutionary patterns and potential functions of CsANNs, we used the neighbor-joining (NJ) method to construct a phylogenetic tree based on 111 ANN proteins from nine species. All the ANNs could be divided into five subfamilies according to the phylogenetic tree analysis (Figure 2). CsANNs were distributed among all subfamilies except for subfamily III and V, and their distribution patterns were similar with other ANNs except three in *C. yanhusuo* (CyANN16, CyANN24 and CyANN38) (Figure 2). Among all nine CsANNs, CsANN1, 7, and 9 were classified within subfamily I, CsANN2, 3, 4, 6, and 8 were assigned to subfamily IV, and only CsANN5 was categorized under subfamily II (Figure 2). Typically, CsANNs were initially grouped with CtANNs and CyANNs, followed by PtANNs and AtANNs within each clade, suggesting a closer evolutionary relationship among CtANNs, CyANNs, and CsANNs (Figure 2). Conversely, the evolutionary relationships among the ANN proteins from *C. saxicola*, *Hordeum vulgare*, and *O. sativa* were found to be the furthest away (Figure 2). These findings indicated that CsANNs were more closely related to the one found in dicotyledons when compared to those of monocotyledons, which was consistent with previous studies [18,21,22,23].

### 2.3. Conserved Motifs and Gene Structure of CsANNs

The gene structure characteristics of *CsANN*s and the conserved motifs and domain compositions of CsANN proteins were analyzed, and they are shown according to their phylogenetic relationships (Figure 3A). Four or five motifs were characterized in CsANNs, and motifs 1~4 were commonly identified in all CsANNs (Figure 3B). CsANN2 and CsANN8, two members of subfamily IV, lacked motif 5, potentially causing functional loss (Figure 3B). Additionally, all CsANNs have four ANN domains except CsANN2 and 8, which lack one and two ANN domains, respectively (Figure 3C). The exon–intron structural analysis showed that the number of exons within the *CsANN* genes varied between 4 and 7 (Figure 3D). Notably, *CsANN3* contained 7 exons, and *CsANN8* contained 4 exons (Figure 3D). It should be mentioned that among all *CsANN*s, *CsANN1* possesses the longest introns (Figure 3D). Furthermore, *CsANN*s within the same evolutionary branches, specifically subfamily I (*CsANN1* and *CsANN9*) and subfamily IV (*CsANN4* and *CsANN6*), exhibited similar intron–exon patterns (Figure 3D).

### 2.4. Chromosomal Location and Collinearity Analysis of CsANNs

A chromosomal location map was constructed to investigate the genetic divergence and duplication within the *CsANN*s. Nine *CsANN* genes were identified and they were localized on five chromosomes. The distribution of *CsANN*s on each chromosome appeared relatively independent and irregular. Among these chromosomes, chromosome 8 contained the highest number of *CsANN*s, with three identified, followed by chromosomes 6 and 7, each with two *CsANN*s. Chromosomes 1 and 2 each contained a single *CsANN* gene (Figure 4A).

To further elucidate the expansion mechanism of the *CsANN* gene family, we performed collinearity and duplication analyses using the TBtools v2.096 [34]. Intraspecific collinearity analysis revealed the presence of one pair of collinear genes in *C. saxicola* (*CsANN1*-*CsANN9*), and both were attributed to whole genome duplications (WGDs), indicating that *CsANN1* and *CsANN9* were orthologous genes. Such an event might bring about the subsequent sub-functionalization of *CsANNs* and facilitate an increase in gene function complexity. *CsANN3*, *4*, *5*, and *6* arose from tandem repetitions (Figure 4B), which might facilitate functional differentiation and the division of function through the generation of sequence variations. These indicated that WGD and tandem repetition events were pivotal for the expansion of *CsANNs* during evolution. Subsequently, the expansion of the *CsANN* gene family might enhance the adaptive potential of *C. saxicola* in response to environmental stress in karst regions.

To explore the underlying evolutionary mechanisms of *CsANN*s, we selected five representative angiosperm species, including *A. thaliana*, *C. tomentella*, *H. vulgare*, *O. sativa*, and *P. tremula*, to construct collinearity analysis maps with *C. saxicola*. The analysis revealed that the *CsANN*s exhibited the highest syntenic relationships with ANNs in *P. tremula* (10), followed by *C. tomentella* (8), *A. thaliana* (5), *H. vulgare* (4), and *O. sativa* (3) (Figure 5). These results indicated that the *ANN* genes from *C. saxicola*, *P. tremula*, and *C. tomentella* had a close relationship. In addition, a greater number of *ANN* homologous genes were identified in dicotyledons when compared to those in monocotyledons (Figure 5).

### 2.5. Analysis of Cis-Acting Elements in CsANN Promoters

To further investigate the response of *CsANN*s to stresses, we used PlantCARE to extract the cis-acting elements involved in plant hormone regulation and abiotic stress responses from 2.0 kb 5′ upstream regions of *CsANN*s [35]. As shown in Figure 6, five main categories of cis-acting elements were predicted, including oxidative stress- (ARE-motif), cold stress- (LTR-motif), drought stress- (MBS-motif), light- (Box 4, GATA-motif, G-Box, G-Box, GT1-motif, TCT-motif), and plant hormone- (abscisic acid response element, ABRE; methyl jasmonate response element, CGTCA-motif, TGACG-motif; gibberellin response element, GARE-motif, P-box; auxin response element, TGA-element) responsive elements. All the *CsANN*s, except *CsANN4*, contain light-responsive elements. Cis-acting elements associated with hormonal responses were also found in all *CsANN* promoters, except *CsANN1*. *CsANN1*, *7*, and *4* contained elements related to drought response. *CsANN4*, *7*, and *9* contained elements related to low temperature response. Only *CsANN7* promoters were identified as having a role in the antioxidant response. In addition, the GTGGC-motif was only present in *CsANN9*. The TGACG-motif, which was suggested to be involved in the methyl jasmonate response, appeared only in *CsANN3*. The P-box, one of the gibberellin response elements, was present in the promoters of *CsANN4* and *CsANN9*. The GARE-motif was present in the promoters of *CsANN8* and *CsANN9*. It should be noted that all *CsANN*s had MYB transcription-factor-binding sites (Appendix A). All the observations indicated that the *CsANN* genes might be involved in environmental stress response and hormone regulation, and play essential roles in the physiological and developmental processes of *C. saxicola*.

### 2.6. Tissue-Specific Expression Profile of CsANNs

The qRT-PCR analysis indicated that the expression patterns of the *CsANN*s were variable in different tissues of *C. saxicola*. As shown in Figure 7, *CsANN2*, *CsANN3*, *CsANN4*, and *CsANN8* were highly expressed in flowers, suggesting their potentially significant biological roles in flowering phase transition or flowering organ development. *CsANN5* and *CsANN6* were particularly highly expressed in the stems, indicating that they might be involved in the development of the stems of *C. saxicola*. We also observed that some *CsANN* genes (*CsANN2*, *CsANN7*, and *CsANN8*) displayed high expression levels in fruit pods of *C. saxicola*. Interestingly, *CsANN1*, *CsANN3*, *CsANN4*, and *CsANN9* showed relatively high accumulation in the roots of *C. saxicola*.

### 2.7. The Effects of Exogenous CaCl_2_ Treatments on C. saxicola Seedlings

In order to investigate whether Ca^2+^ influences the growth and development of *C. saxicola*, different concentrations of exogenous CaCl_2_ were used to treat their one-month-old seedlings. As shown in Figure 8A–C, the calcium, proline, and soluble sugar levels of *C. saxicola* leaves were significantly increased when they were treated with increasing concentrations of exogenous CaCl_2_ concentrations (*p* < 0.05). In addition, significant differences in the DHCA levels were also observed in the stems, leaves, and roots of *C. saxicola* after the treatments (*p* < 0.05), and with much higher amounts found in the roots than in the stems and leaves (Figure 8D–F). Interestingly, high concentrations of exogenous CaCl_2_ also significantly induced the accumulation of DHCA in the roots of *C. saxicola* (Figure 8F).

We then analyzed the Fragments Per Kilobase Million (FPKM) values of *CsANN*s based on the transcriptomic data. It was found that, except for *CsANN2* and *CsANN8*, the FPKM values of other *CsANN*s showed significant changes as the concentrations of CaCl_2_ increased (*p* < 0.05) (Figure 8G–O). In particular, the FPKM values of *CsANN1* and *CsANN9* were positively correlated with calcium concentrations (Figure 8G,O).

### 2.8. CsANNs Correlated with DHCA Biosynthesis

According to the predicted biosynthetic pathway of DHCA, C-methyltransferases (CMTs), O-methyltransferases (OMTs), and berberine bridge enzyme-likes (BBELs) were involved in the transition of cheilanthifoline into DHCA [36]. In order to elucidate the relationship between CsANNs and the above key enzymes involved in DHCA biosynthesis, we conducted a correlation analysis among *CsCMT*s, *CsOMT*s, *CsBBEL*s, and *CsANN*s. An interaction network comprising 34 nodes and 66 relational pairs with high correlations (R ≥ 0.8 and *p* ≤ 0.05 or R ≤ −0.8 and *p* ≤ 0.05) was identified. Among them, four major sub-networks (*CsANN1*, *CsANN5*, *CsANN6*, and *CsANN9*), which collectively included 48 relational pairs and 25 nodes, were summarized (Appendix A). For *CsANN1*, the connectivity number was nine, and this showed a significant correlation with the expression of three *CsOMT*s and six *CsBBEL*s. Similarly, the connectivity number for *CsANN9* was 11, which was significantly correlated with the expression levels of five *CsOMT*s and six *CsBBELs* (Appendix A).

As is mentioned above, the transcripts of both *CsANN1* and *CsANN9* were not only relatively high in the roots of *C. saxicola*, but also correlated positively with the levels of DHCA in the roots of *C. saxicola* treated by exogenous CaCl_2_ solutions. Furthermore, the expression of *CsANN1* and *CsANN9* were highly correlated with the expression of DHCA-biosynthetic-related genes (Appendix A). These indicated that CsANN1 and CsANN9 might play positive roles in the biosynthesis of DHCA.

To confirm this speculation, transient overexpression assays of *CsANN1* and *CsANN9* were conducted in *C. saxicola* leaves, respectively (Figure 9A). The transcript levels of *CsANN1* and *CsANN9* in *C. saxicola* leaves that were infiltrated with *CsANN1*-pCAMBIA1301 and *CsANN9*-pCAMBIA1301 increased significantly when compared to the empty vector controls (EV), respectively (Figure 9B and Appendix A). Furthermore, the concentrations of cheilanthifoline and DHCA in *C. saxicola* leaves infiltrated with *CsANN1*-pCAMBIA1301 were also significantly increased when compared to EV (Figure 9C,D). However, compared to EV, the contents of cheilanthifoline and DHCA in *C. saxicola* leaves infiltrated with *CsANN9*-pCAMBIA1301 did not show significant differences (Appendix A). These results suggested that CsANN1 might play a positive role in the biosynthesis of DHCA.

### 2.9. Yeast Two Hybrid Assays and Protein Interaction Networks of the CsANNs

Numerous proteins can form dimers, polymers, and intricate complexes to execute their biological processes in vivo [37]. In order to explore whether CsANNs could form homodimers and heterodimers, yeast two-hybrid assays were conducted. As shown in Figure 10A, similar to the negative control pGBKT7 (pBD), the yeast cells carrying the tested vectors could not grow on the selective solid media that lacked tyrosine (Trp) but contained 5-bromo-4-chloro-3-indoxyl-α-D-galactopyranoside (X-α-gal) and aureobasidin A (AbA). This indicated that all CsANNs had no auto-activation activities in Y2HGold cells (Figure 10A). Further assays indicated that neither of the yeast cells containing the pair of co-transformed indicated plasmids could grow on the selective medium containing X-α-gal and AbA, indicating that none of these CsANNs could form homodimers or heterodimers in yeast cells (Figure 10B).

We then utilized the STRING database and TBtools v2.096 to reveal the interactions between CsANNs and other proteins. The results showed that the interaction patterns of CsANN1 with proteins closely resemble those of AtANN1 (Figure 10C). The proteins that interacted with AtANN1 included the DEAD-box ATP-dependent RNA helicase (AtRH11, 37, 52) and it belonged to the DDX3/DED1 subfamily of the DEAD-box helicase family. This was probably a cyclic nucleotide-gated ion channel (AtCNGC1, 6), which belonged to the cyclic nucleotide-gated cation channel. The tetraspanin-8 (AtTET8) belonged to the tetraspanin family. The hyperosmolarity-gated non-selective cation channel (AtOSCA1) belonged to the CSC1 family. Similarly, the interaction patterns of CsANN9 with proteins were similar to those of AtANN2. AtANN2 was predicted to interact with AtTET8; RGG repeats nuclear RNA-binding protein C (AtRGGC); RGG repeats nuclear RNA-binding protein B (AtRGGB), which belongs to the RGGA protein family; UDP-glucose 6-dehydrogenase 4 (AtUGD4) belongs to the UDP-glucose/GDP-mannose dehydrogenase family, and other proteins. Furthermore, the interaction network of CsANNs suggested potential interactions with proteins such as glucose-6-phosphate isomerase (GPI), GDP-mannose transporter GONST1 (GDP-Man), and ATP-dependent RNA helicase DDX3X (DDX3X), among others (Figure 10C). These results indicated that the CsANN proteins might be involved in various physiological functions, including transcriptional regulation, RNA stability, signal transduction, mRNA transport, Golgi secretion, the maintenance of cellular osmotic balance, the synthesis of cell-wall precursors, and the regulation of the calcium channels.

## 3. Discussion

### 3.1. The Structure of CsANNs Were Conserved

The ANNs have been identified and they were functionally analyzed in a variety of plant species, including model plants and crops, but the information regarding the *ANN* gene families in medicinal plants is relatively scarce [19,20,21,22,23]. In this study, nine *CsANN* genes were identified based on the genomic and transcriptomic data of *C. saxicola*, an endangered herbaceous plant that exclusively inhabits in Chinese karst landforms and it is widely used in traditional Chinese medicine. All the CsANN proteins have four typical ANN domains, except for CsANN2 and CsANN8, which lack one and two ANN domains, respectively (Figure 3C). It is well known that a highly conserved sequence (G/KXGT-38-D/E) for binding sites of type II Ca^2+^ is present in each ANN domain in vertebrates [10]. However, the type II Ca^2+^-binding sites are absent in the repeats II and III in plant ANNs [9]. Our results also showed that repeats II and III of CsANNs did not contain the type II Ca^2+^-binding sites, which was in accordance with other plant ANNs [9,38,39]. In addition, repeat I was absent in CsANN2 and CsANN8, and repeat IV was absent in CsANN8 (Figure 1B). Furthermore, although four repeats were all present in the other seven CsANNs, the type II Ca^2+^-binding sites were less conserved, with two or three substitutions in repeat IV of CsANNs. Actually, the type II Ca^2+^-binding sites were only highly conserved in repeat I of CsANN1, 5, 7, and 9 (Figure 1B). Those absence of type II Ca^2+^-binding sites or the entire repeats might result in different protein conformations, and these changes would likely weaken the ability of the CsANN proteins to bind to the phospholipids of cell membranes in a Ca^2+^-dependent manner, further influencing their roles in plant growth and development regulation [13,15,20].

Another motif present in repeat I was the ILAHR sequence (Figure 1B). It was demonstrated that this sequence, especially the conserved His40 residue, played an essential role in peroxidase activity [40,41]. For example, *Arabidopsis* AtANN1 has shown the His-residue-dependent peroxidase activity in vitro [16]. An BrANN1-contained complex protein purified from *B. rapa* floral buds had peroxidase activity [38]. A maize ANN preparation also demonstrated peroxidase activity that appeared independent of heme association [40]. It is believed that the peroxidase activity of ANNs could protect the membrane from peroxidation and then subsequently improve the abiotic tolerance in plants, such as drought tolerance [12,42]. As plants such as *C. saxicola* in karst regions experienced harsh environmental conditions, the CsANNs (CsANN1, 5, 9), which present the ILAHR sequences, might contribute calcareous and drought tolerance abilities to *C. saxicola* plants partly by the peroxidase activities. Further enzymatic assays are needed to confirm this activity in *C. saxicola*.

The IRI sequence in repeat III of ANNs was highly conserved in CsANN1 and 9, and with one or two substitutions in the other CsANNs, except for CsANN2 (Figure 1B). The IRIs present in ANNs were demonstrated to bind F-actin but not globular actin in a Ca^2+^-dependent manner, indicating these ANNs might be involved in the regulation of exocytosis [43]. Moreover, several ANNs of maize, cotton, and tomato were reported to hydrolyze ATP and GTP, and the nucleotide hydroxylation abilities were attributed to the Walker A motif (GXXXXGKT/S) and the GTP-binding domain (GXXG), which were both retained in repeat IV of ANNs [9,13]. CsANN1, 5, and 9 contain a modified Walker A motif with one or two substitutions and the conserved GXXG sequence in the repeat IV, indicating their potential abilities to bind and hydrolyze GTP/ATP [11,14,15,16,17]. Interestingly, similar to *A. thaliana*, cotton, maize, and poplar, the nucleotide-binding domains in these species were partly overlapped with the Ca^2+^-binding site in repeat IV, indicating that the two different binding properties were contradictory to each other, and therefore, they controlled the spatiotemporal functions of ANNs [12,24,25,44].

The phylogenetic analysis showed that CsANNs were distributed in three subfamilies, and they clustered with ANNs from the dicotyledonous plants, especially the ANNs in the same genus (*C. tomentella* and *C. yanhusuo*) (Figure 2). Similarly to other plant ANNs, the closely related CsANNs tend to exhibit similar structural features [20,21,24,39]. As shown in Figure 3B, motifs 1–4 are common to all CsANN members, indicating that it is crucial for the CsANNs. Nevertheless, the members of subfamily II lacked motif 5, and those of subfamily III had significant differences in their exon–intron structures, which may be related to functional differentiation. It is worth noting that *CsANN1* had more exons and longer introns, implying that the gene plays a more crucial role in function, expression regulation, and evolution [45,46].

### 3.2. Involvement of CsANNs in the Biosynthesis of DHCA

It was interesting that the expression profiles of *CsANN1* and *CsANN9* in variable tissues and Ca^2+^-treated roots of *C. saxicola* were similar to the accumulation profiles of DHCA in these treated and untreated organs. This was indicative of the involvement of *CsANN1/9* in DHCA biosynthesis (Figure 7 and Figure 8). In fact, the transient overexpression of *CsANN*s in *C. saxicola* leaves demonstrated that higher expression levels of *CsANN1* could really result in the higher accumulation of BIAs, including palmatine, cheilanthifoline, dehydrocheilanthifoline, and cavidine, when compared to that of EV (Figure 9C,D). In particular, compared to EV, the levels of DHCA and its crucial upstream precursor compound, cheilanthifoline, were significantly elevated in *CsANN1*-overexpressed *C. saxicola* leaves. This indicated that CsANN1 might positively regulate the expression of the genes and proteins involved in the biosynthesis pathway of cheilanthifoline and DHCA (Figure 9D). Further analysis of the candidates that participated in the biosynthetic pathway of DHCA showed that the expression level of *CsBBEL9* transcripts was significantly upregulated in *CsANN1*-overexpressed *C. saxicola* leaves compared to that of EV (Figure 9B). As a result, we speculated that CsANN1 might be indirectly involved in up-regulating *CsBBEL9*, possibly via interaction with transcriptional regulators. In addition, CsANN1 might bind directly to the DNA sequence of *CsBBEL9* in the nucleus, similar to transactivation factors, to regulate the expression of *CsBBEL9*, thereby inducing the accumulation of DHCA and its precursors. Subcellular localization analyses of CsANN1 and CsANN9 suggested a more likely presence in the cytoplasm and cytoskeleton (Figure 1A). However, studies have shown that some ANNs could be introduced into the nucleus through unknown mechanisms, especially under stress conditions [10,47]. One possible method of assistance might be supported by the conformational changes in ANN proteins induced by N-terminal tyrosine phosphorylation, allowing the proteins to enter the nucleus [9,13,47,48]. However, more biochemical and structural evidence should be provided to confirm the subcellular localizations of ANNs, including CsANNs, during physiological processes.

Another factor influencing the accumulation of DHCA by CsANN1 might occur at the translational level. A number of cis-acting elements, including MYB-binding sites, were identified in the promoters of *CsBBEL*s, *CsCMT*s, *CsOMT*s, and *CsANN*s, and the Y2H assays demonstrated that CsANNs could interact with several CsMYBs in Y2HGold yeast cells. Therefore, CsANNs-CsMYBs modules might play crucial roles in the biosynthetic pathway of DHCA. Further studies, such as the knockdown and knockout of *CsANN* and *CsMYB*s by RNA interferences or clustered regularly interspaced short palindromic repeats (CRISPR)/CRISPR-associated protein 9, as well as the overexpressing of *CsANN* or *CsMYB*s, should be conducted to unravel the precise mechanisms of the biosynthesis of DHCA at transcriptional and translational levels. This will help to elucidate the adaptations of medicinal plants in karst regions.

### 3.3. The Role of CsANNs in the Adaptability of C. saxicola to a Karst Environment

It was reported that the calcium content in karst soil could reach up to 37.68 g/kg at most, and the average calcium availability of these soils was as high as 50.9% [32,49]. Preventing excessive absorption of Ca^2+^ is crucial for the normal growth of karst-adapted plants. For this purpose, variable unique physiological mechanisms and responsive pathways have been evolved to maintain the balance of Ca^2+^ within plant cells. This involves fixing excess Ca^2+^ by forming calcium oxalate crystals [50]. The plants also release excess Ca^2+^ through their stomata on mature leaves [51,52] and they regulate the concentration of Ca^2+^ by storing it in the intercellular spaces and organelles, such as vacuoles, endoplasmic reticulum, mitochondria, and chloroplasts [51,53]. To date, several key factors have been demonstrated to be involved in these processes. For example, the Ca^2+^/cation antiporter superfamily, which are classified into four families including H^+^/cation exchangers (CAXs), Na^+^/Ca^2+^ exchanger-like proteins, cation/Ca^2+^ exchangers, and Mg^2+^/H^+^ exchangers, which are all critical for regulating and accumulating calcium [49,51]. Physiological and genetic evidence has demonstrated that CAXs could mainly remove excess Ca^2+^ from the cytoplasm into the vacuole to maintain calcium homeostasis and alleviate calcium-induced stresses, thus granting plants the ability to adapt to the high-calcium of karst environments [51,54]. In addition, Ca^2+^-ATPase (ACA) could also prompt Ca^2+^ efflux from cytosol to endoplasmic reticulum, or facilitate Ca^2+^ uptake from cytosol to vacuole, in an ATP-dependent manner [55]. In our study, compared with those in CK, the expression levels of several *CsCAX*s, *CsACA*s, and other Ca^2+^-balancing genes were significantly upregulated in *C. saxicola* roots treated with high concentrations of CaCl_2_ solution (Appendix A), indicating the involvement of these factors in the adaptation of *C. saxicola* in karst landforms. In addition, the expression levels of some *CsCAX*s were much higher than those of *CsACA*s and other Ca^2+^-balancing genes (Appendix A). This meant that the shaping of the efflux elements of Ca^2+^ signatures in *C. saxicola* led to the expression of *CsCAX*s other than *CsACA*s. This is consistent with the fact that CAXs have lower energy consumption than ACAs and mainly act on the vacuole membrane, and they are more conducive to the secretion and long-term storage of Ca^2+^ [51,56,57]. Interestingly, we identified significant correlations between CsANNs and the Ca^2+^-balancing genes (Appendix A), indicating the complicated adaptation styles of *C. saxicola* in calcium-rich karst ecosystems.

Except for the involvement in the adaptation of high-calcium environment, ANNs could also respond to other stresses to help regulate the growth and development of plants, at least partly attributed to the diversity of their cis-acting elements in their promoter regions [9,11,16]. Multiple cis-acting elements were also identified in the promoters of the *CsANN*s (Figure 6), indicating the role of these genes in the adaptation of *C. saxicola* to the complex ecological environment in karst areas. For example, the ABRE element-binding sites, which could respond to ABA signals and balance cellular osmotic pressure, are present in the promoters of *CsANN1*, *3*, *4*, and *9* [58]. All *CsANN*s possess the cis-elements of MYB transcription factors, which indicated their involvements in drought resistance by regulating ion homeostasis and reducing water loss [59]. In addition, the antioxidant response elements could activate antioxidant enzymes, such as superoxide dismutase and catalase, by binding to the nuclear factor-erythroid 2-related factor 2 or basic (region-leucine) zipper transcription factors. This would reduce the damage caused by reactive oxygen species produced by adverse environmental factors to the membrane lipids of plants, present in all *CsANN*s except *CsANN7* [60]. Furthermore, the *CsANNs’* promoters also contain a variety of hormone-responsive elements, such as CGTCA-motif (cis-acting regulatory element involved in the MeJA reaction), TGACG-motif (methyl jasmonate response), P-box (gibberellin response), and TGA-element (auxin response) (Appendix A). All of these are conducive to alleviating stresses to the growth, development, and differentiation of plants caused by arid, infertile, and heavy metal pollutions [61,62,63]. Based on these, we speculated that CsANNs may alter the concentrations of free Ca^2+^ in the cytoplasm by interacting with multiple signaling pathway genes. As Ca^2+^ is a second messenger in plants, the characteristic changes of [Ca^2+^]_cyt_ in different environments are recognized and decoded as regulatory responses in DNA, RNA, protein, and metabolic levels to help *C. saxicola* adapt in harsh karst environments.

## 4. Materials and Methods

### 4.1. Plant Materials and Treatments

*C. saxicola* seeds were germinated and grown in pots containing a matrix with a 3:1 mixture ratio of peat soil and vermiculite in the greenhouse (23 °C, 16 h light, a photo flux density of 120 µmol m^−2^ s^−1^). For the exogenous CaCl_2_ treatment experiments, the one-month-old *C. saxicola* plants were treated with the equal volume of CaCl_2_ at 4 mmol/L, 30 mmol/L, 100 mmol/L, 200 mmol/L, 300 mmol/L, and 400 mmol/L, respectively, for 25 days. The same batch of plants treated with the equal volume of water were set as the controls (CK). The organs were then cleaned, physically isolated, and they were immediately frozen at −80 °C or dried for subsequent studies. All samples for each treatment and CK were performed in three biological replicates.

### 4.2. RNA Sequencing and Transcriptomic Data Analysis

Total RNA was extracted from seven tissues of *C. saxicola*, in addition to the roots of *C. saxicola* treated with different concentrations of CaCl_2_, employing the pBIOZOL Plant Total RNA Extraction Reagent (Bioer Technology, Hangzhou, China). RNA concentration and integrity were assessed by using a Nanodrop 2000 spectrophotometer (Thermo Scientific, Wilmington, DE, USA) and a Qubit 3.0 Fluorometer (Life Technologies, Carlsbad, CA, USA), respectively. The cDNA libraries were prepared using the NEB Ultra II RNA Library Prep Kit (New England Biolabs, Boston, MA, USA) according to the manufacturer’s instructions. RNA sequencing was performed on the Illumina NovaSeq 6000 platform (Illumina, San Diego, CA, USA). The raw reads were quality controlled and cleaned using SOAPnuke v1.5.6 [64]. Trimmed sequences were aligned to the reference genome using HISAT2 v2.2.1 [65]. Gene expression was normalized as the FPKM with Stringtie v2.1.4 [66].

### 4.3. Genome-Wide Identification of the CsANN Genes

The genome data of *A. thaliana* (https://www.ncbi.nlm.nih.gov/datasets/genome/GCF_000001735.4/; GCA_000001735.2; accessed on 19 February 2025) and *O. sativa* (https://www.ncbi.nlm.nih.gov/datasets/genome/GCF_034140825.1/; GCA_034140825.1; accessed on: 19 February 2025) were downloaded from the NCBI, and those of *B. oleracea* (https://ftp.ebi.ac.uk/ensemblgenomes/pub/release-61/plants/fasta/brassica_oleracea/dna/; GCA_000695525.1; accessed on 20 February 2025), *B. rapa* (https://ftp.ebi.ac.uk/ensemblgenomes/pub/release-61/plants/fasta/brassica_rapa/dna/; GCF_034140825.1; accessed on 20 February 2025), and *H. vulgare* ((https://ftp.ebi.ac.uk/ensemblgenomes/pub/release-61/plants/fasta/hordeum_vulgare/dna/; GCF_034140825.1; accessed on 19 February 2025) were downloaded from the EnsemblPlants. The genome of *P. tremula* was downloaded from the phytochrome (https://phytozome-next.jgi.doe.gov/info/PtremulaxPopulusalbaHAP1_v5_1; phytozome genome ID: 717; accessed on 23 February 2025), that of *C. yanhusuo* was from the CNGB Sequence Archive (CNSA) (https://db.cngb.org/search/project/CNP0004356/; CNP0004356; accessed on 3 March 2025), and *C. tomentella* was downloaded from the Genome Warehouse in the National Genomics Data Center (https://ngdc.cncb.ac.cn/gwh/Assembly/10358/show; GWHAORS00000000; accessed on 7 March 2025). BGI Genomics, Shenzhen, China, conducted the de novo assembly of the *C. saxicoia* genome (https://www.ncbi.nlm.nih.gov/datasets/genome/GCA_047716255.1/; accessed on 1 April 2025). The hidden Markov model (HMM) with the canonical ANN domain (Pfam00191) was used to search against the *C. saxicola* genome dataset (*p* < 0.001), which was assembled by us, to identify *CsANN* genes [67]. Then, all eight *AtANN*s downloaded from The Arabidopsis Information Resource (TAIR) database were used as queries to retrieve *CsANN*s by BLAST v2.2.25 in TBtools v2.096 [34]. Lastly, the putative *CsANN*s were subsequently confirmed by using the InterPro [68] and SMART databases [69]. The above non-redundant sequences were considered as *CsANN* candidates. The protein physicochemical properties of CsANNs were analyzed using the online website, ExPASy (https://web.expasy.org/protparam/; accessed on 23 February 2025) [70]. Subcellular localizations of CsANNs were predicted using the online websites, Wolf-psort (https://wolfpsort.hgc.jp/) [71], Cell-PLoc 2.0 (http://www.csbio.sjtu.edu.cn/bioinf/Cell-PLoc-2/; accessed on 24 February 2025) [72], CELLO (http://cello.life.nctu.edu.tw/; accessed on 23 February 2025), PSORT2 (https://www.genscript.com/psort.html; accessed on 23 February 2025) [73], and Euk-mPLoc 2.0 (http://www.csbio.sjtu.edu.cn/bioinf/euk-multi-2/; accessed on 25 February 2025) [74].

### 4.4. Phylogenetic Relationship, Conserved Motif, and Gene Structure Analysis

The multiplication alignment of the CsANNs was performed by the TBtools v2.096 [30]. The results were then visualized using the Jalview 9.0.5 [75]. The ANN protein sequences of *A. thaliana*, *B. oleracea*, *B. rapa*, *C. saxicola*, *C. tomentella*, *C. yanhusuo*, *H. vulgare*, *O. sativa*, and *P. tremula* were aligned using MEGA 11.0 [14]. The phylogenetic tree of the ANNs was constructed using the neighbor-joining method, with a bootstrap value of 1000 replicates. The analysis was conducted with the amino acid substitution type, pairwise deletion for gap data treatment, and the Poisson model, while the other parameters were maintained at their default settings. The resulting tree was subsequently visualized using FigTree v1.4.4. The conserved motifs of CsANNs were identified using the MEME Suite 5.5.7, with the maximum number of motifs being set to 10 [76]. The intron and exon regions of *CsANN*s were analyzed based on the genomic data of *CsANN* DNA using the visualize gene structure program in TBtools v2.096 [34].

### 4.5. Chromosomal Location, Collinearity, and Gene Duplication Analysis

The chromosome localization information of the *CsANN* genes was extracted from the *C. saxicola* plant genome gff file by using the TBtools v2.096 software, and visualization images were generated [34]. TBtools v2.096 was further employed to conduct and visualize the collinearity analysis of the ANNs. The default value of minimum match score was 50, and the maximum gap was 20 [34]. Tandem duplication events are characterized by the presence of two or more contiguous genes on the same chromosome. WGD events were identified by the presence of duplicate gene pairs, which were situated on different chromosomes [15]. The gene duplication analysis was also determined by TBtools v2.096 [34].

### 4.6. Cis-Acting Elements, and Protein–Protein Interaction Analysis

The 2 kb sequences upstream of the 5′ UTR of *CsANN* genes were extracted using the TBtools v2.096 software and then used for the prediction of cis-acting elements based on the PlantCARE database [35]. Protein–protein interactions were predicted by using the STRING database [77]. Cytoscape V3.9 was used for visualization of the results [78].

### 4.7. Measurement of the BIAs, Calcium, Proline, and Soluble Sugar Contents

An Agilent 1260 Infinity II (Agilent Co., Ltd., Santa Clara, NY, USA) was employed to determine the relative levels of cavidine, (s)-cheilanthifoline, dehydrocheilanthifoline, DHCA, and palmatine. Initially, 200.0 mg of dried *C. saxicola* samples were dissolved in 10 mL of methanol and then ultra-sonicated for 60 min. Next, the samples were centrifuged at 13,000× *g* for 15 min, and the supernatants were removed to new 15 mL tubes and dried using a nitrogen pressure reduction method at room temperature. Thereafter, the samples were dissolved in 1 mL of the mobile phase (acetonitrile: 0.01% K_2_HPO_4_ aqueous solution = 21:79 (*v*/*v*)) and filtered through a 0.22-μm filter prior to analysis. The Agilent XDBC 18 chromatographic column (150 mm × 4.6 mm, 5 µm) was used to elute the BIAs in samples and these compounds were detected at a wavelength of 347 nm at 30 °C. The volumetric flow rate of the mobile phase (acetonitrile: 0.01% of K_2_HPO_4_ = 21:79 (*v*/*v*)) was 1.0 mL/min.

The calcium concentrations were quantified using an atomic absorption spectrophotometer (Model 5100 PC, Perkin-Elmer Corporation, Waltham, MA, USA) [79]. The soluble sugar content was assessed by using the anthrone reagent method [80]. Free proline levels were determined through a ninhydrin colorimetric assay, following extraction with yellow salicylic acid [81]. Each reaction was conducted in triplicate.

### 4.8. CsANN Cloning

RNA was extracted from the leaves, stems, and roots of *C. saxicola* before and after various treatments by FastPure Universal Plant Total RNA Isolation kit (Vazyme, Nanjing, China) based on the manufacturer’s instruction. The first-strand cDNA was synthesized according to the instruction of HiScript III 1st Strand cDNA Synthesis Kit (Vazyme, Nanjing, China) using 1 μg of the extracted total RNA as the template. To clone the full-length CDSs of *CsANN*s, gene-specific primers (Appendix A) were designed. The PCR reaction procedures were as follows: pre-denaturation at 95 °C for 3 min; followed by 33 cycles of denaturation at 95 °C for 15 s, annealing at 56 °C (*CsANN2*, *6*) or 54 °C (other *CsANN*s) for 15 s, and extension at 72 °C for 50 s; and a final extension at 72 °C for 5 min. Subsequently, PCR products were ligated into the TA/Blunt-Zero Cloning Kit vector (Vazyme, Nanjing, China) and then transformed into DH5α competent cells for incubation on LB solid medium plus 100 µg mL^−1^ kanamycin overnight. Colony PCR was performed using the same procedure as above, and this was conducted to verify the putative positive transformants. The final CDSs of *CsANN*s were ascertained using Sanger sequencing.

### 4.9. qRT-PCR Analysis

The first-strand reaction products were diluted with sterilized triple-distilled H_2_O, and the diluted products (1 µL) were used for qRT-PCR (20 µL). The assay was performed using ChamQ Universal SYBR qPCR Master Mix (Vazyme, Nanjing, China) on a real-time PCR instrument (Thermo Fisher Scientific Quant Studio^TM^ 3, Waltham, MA, USA). The relative expression levels of specific genes were calculated using the 2 delta delta Ct method (2^−△△Dt^ method), with *glyceraldehyde-3-phosphate dehydrogenase 8* (*CsGAPDH8*) used as the internal control [82]. Before conducting qRT-PCR, the best primer was screened from three pairs in order to confirm the amplification quality and the correctness of the target gene. The qRT-PCR primers are listed in Appendix A. Each reaction was performed in triplicate.

### 4.10. Yeast Two-Hybrid Assay

The full-length CDS of *CsANN*s were amplified and cloned into the destination vectors pGADT7 (pAD) and pBD, respectively. The primers used are listed in Appendix A. Transactivation analysis assays were carried out in the yeast strain Y2HGold using the Yeastmaker Yeast Transformation System 2 (Takara, Tokyo, Japan). For Y2H assays. *CsANN*-pAD and *CsANN*-pBD were transformed into Y187 and Y2HGold yeast cells, respectively. Yeast mating was conducted and the putative co-transformants were screened on SD/-Trp/-Leu (DDO) selective solid medium and then verified by PCR. SD/-Ade/-Trp/-Leu (TDO)+X+A and SD/-His/-Ade/-Trp/-Leu (QDO)+X+A were further adopted to verify the protein–protein interactions. The co-transformation of plasmids, pBD-53 and pAD-T as well as pBD-Lam and pAD-T, were used as positive and negative controls, respectively.

### 4.11. Transient Overexpression of CsANN1 and CsANN9 in C. saxicola Leaves

The full-length CDS of *CsANN1* and *CsANN9* were amplified and then they were cloned into the pCAMBIA1301 vector, respectively. The primers used are listed in Appendix A. The resulting plasmids were introduced into *Agrobacterium tumefaciens* GV3101. After propagation, the cells were re-suspended in infiltration buffer (composed of 10 mmol/L MgCl_2_, 10 mmol/L 2-morpholinoethanesulfonic acid, and 150 mmol/L acetosyringone; pH 5.6) to achieve an OD_600_ of 0.8. Suspensions containing either the overexpressing constructs of the target genes *CsANN1* and *CsANN9,* or the empty pCAMBIA1301 plasmid vector (EV, serving as a negative control), were injected into opposite sides of the same leaf. Two days later, the injected leaves of *C. saxicola* were collected for metabolite content and gene expression analysis using HPLC and qRT-PCR, respectively. Each treatment was conducted in triplicate.

### 4.12. Statistical Analysis

The experimental data were analyzed utilizing Excel 2021 and IBM SPSS Statistics version 22, while Excel was employed for graph plotting. A one-way analysis of variance (ANOVA) was conducted to assess significant differences (*p* < 0.05) among variable treatments. Gene correlation analysis was conducted based on the Pearson analysis using OmicShare (https://www.omicshare.com/tools/; accessed on 27 April 2025). The genes with high correlation (R ≥ 0.8 and *p* ≤ 0.05 or R ≤ −0.8 and *p* ≤ 0.05) were screened. The results were visualized as heatmap and network graph images by using TBtools v2.096 and Cytoscape 3.9.1, respectively [78].

## 5. Conclusions

In this study, we first provided a systematic analysis of the *CsANN* gene family in *C. saxicola*. The phylogenetic analysis of ANNs within nine species indicated that all nine CsANNs can be classified into three different subfamilies, with closely related members sharing structural similarities. In addition, various genomic and proteomic features, including gene structures, conserved motifs, chromosomal distributions, collinearity, protein interactions, and cis-acting elements, were characterized systematically. Expression pattern analysis unraveled two candidates, CsANN1 and CsANN9, as the putative positive regulators involved in the biosynthesis of DHCA in *C. saxicola*. Further experiments indicated that the transient overexpression of *CsANN1* could increase the levels of DHCA in *C. saxicola* leaves. This study provides a theoretical basis for understanding the biological functions of the CsANNs in secondary metabolites and adaptation to the adverse stresses of *C. saxicola*.

## Figures and Tables

**Figure 1 plants-14-01974-f001:**
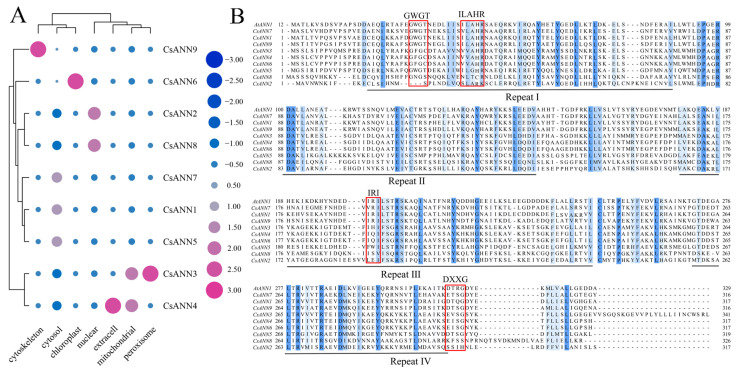
The heatmap of the subcellular localizations of CsANNs ((navy blue signifies absence, blue suggests minimal distribution presence, dark pink suggests distribution presence, and pink indicates significantly greater distribution in the specific region) (**A**)) and multiple sequence alignments of deduced amino acid sequences of ANN proteins from *Corydalis saxicola* Bunting and *Arabidopsis thaliana* (**B**).

**Figure 2 plants-14-01974-f002:**
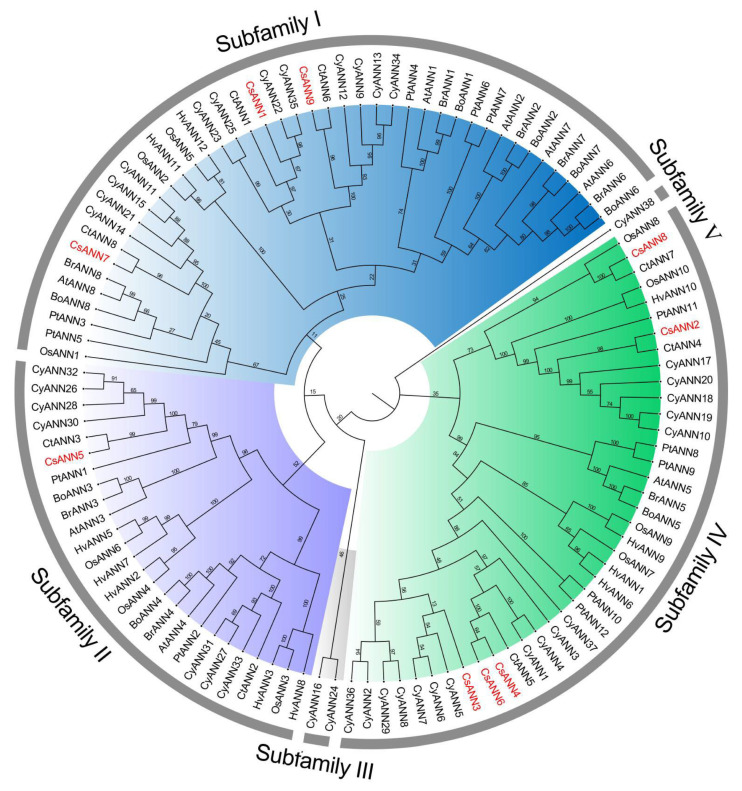
The phylogenetic tree of the ANN proteins. The phylogenetic tree was constructed using the neighbor-joining method tree with 1000 bootstrap replicates. The numbers at the nodes indicate the bootstrap values from 1000 replicates. The red font is used to denote CsANNs. At: *A. thaliana*; Bo: *Brassica oleracea*; Br: *B. rapa*; Ct: *C. tomentella*; Cy: *C. yanhusuo*; Hv: *Hordeum vulgare*; Os: *Oryza sativa*; Pt: *Populus tremula* × Populus.

**Figure 3 plants-14-01974-f003:**
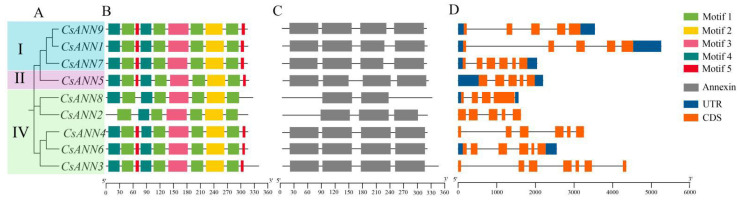
The phylogenetic relationship (**A**), conserved motifs (**B**), ANN domains of CsANN proteins (**C**), and the intron and exon compositions (**D**) of *CsANN* genes. The phylogenetic tree was constructed using the neighbor-joining method tree with 1000 bootstrap replicates. The conserved motifs of CsANNs were identified using the MEME Suite 5.5.7, with the maximum number of motifs being set to 10. The intron and exon regions of *CsANN*s were analyzed using the visualize gene structure program in TBtools v2.096. In the gene structures, the dark blue boxes represent the untranslated regions (UTRs), with the orange boxes representing exons and the black lines representing introns. I, II and IV represent the subfamilies of ANNs displayed in Figure 2.

**Figure 4 plants-14-01974-f004:**
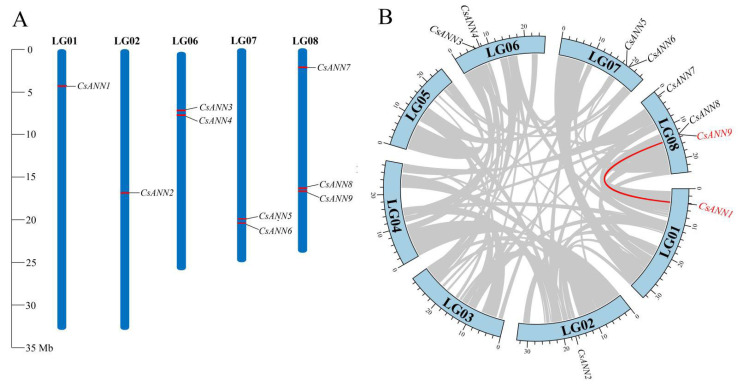
The localized distribution of *CsANN* genes on the *C. saxicola* chromosomes (**A**) and synteny analysis of interchromosomal relationships between *CsANN* genes (**B**). Nine *CsANN* genes were distributed on five of eight chromosomes. For synteny analysis of *CsANN* genes, red lines and gray ones represented the collinear gene pair and the syntenic blocks, respectively.

**Figure 5 plants-14-01974-f005:**
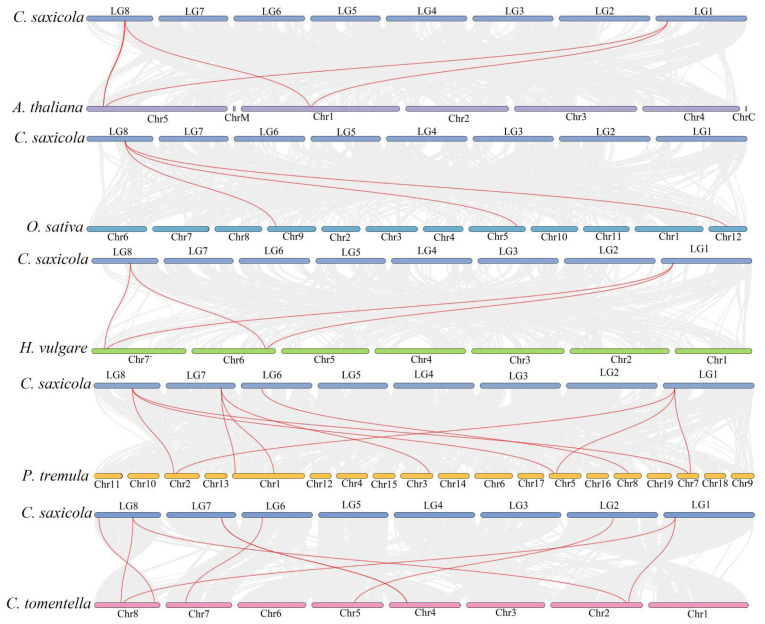
The extra-genomic collinearity related to *CsANN* genes in *A. thaliana*, *C. saxicola*, *C. tomentella*, *H. vulgare*, *O. sativa*, and *P. tremula*. Red lines and gray ones represented the collinear gene pairs and the syntenic blocks, respectively.

**Figure 6 plants-14-01974-f006:**
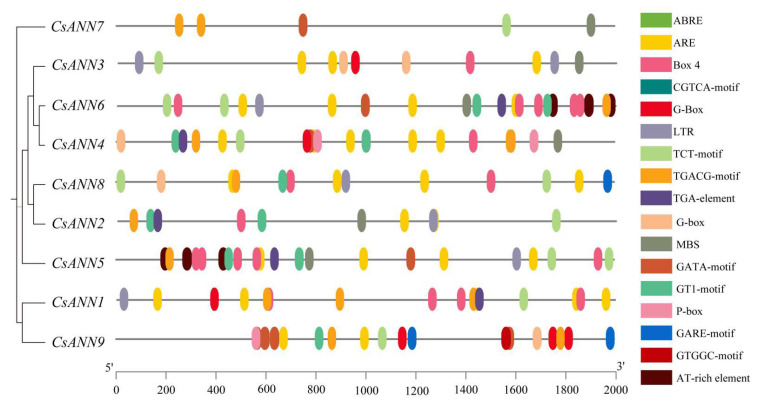
The distribution of cis-acting elements in the promoters of *CsANN*s. The promoter region was defined as a 2.0 kb sequence upstream of the translation initiation codon of the *CsANN* gene. Different types of cis-acting elements in the promoters are represented by ellipses of different colors.

**Figure 7 plants-14-01974-f007:**
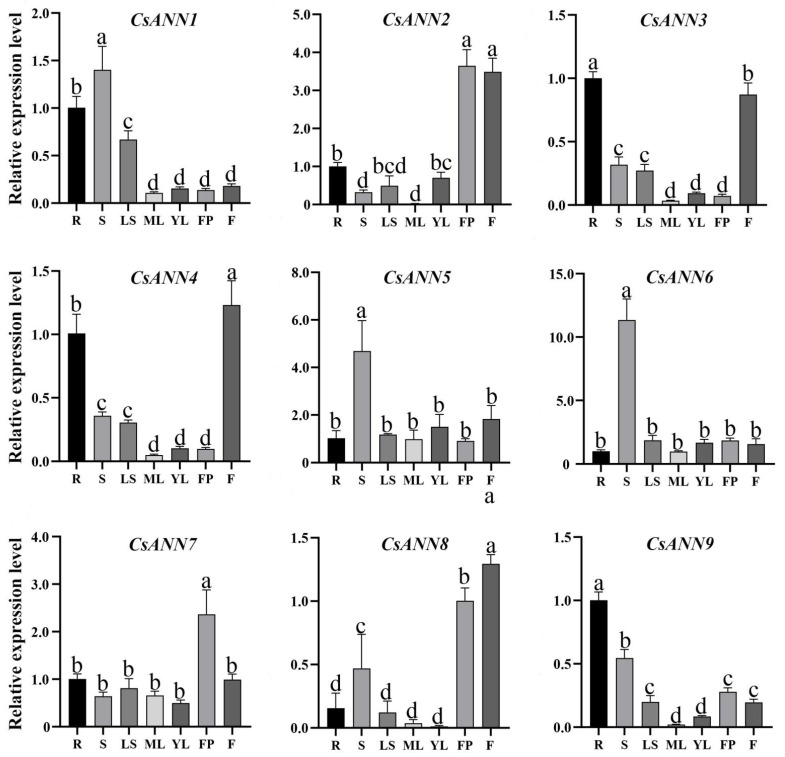
The relative expression levels of *CsANN*s in different tissues of *C. saxicola.* R: roots; S: stems; LS: lateral stems; ML: mature leaves; YL: young leaves; FP: fruit pods; F: flowers. The a, b, c and d represent statistically significant differences.

**Figure 8 plants-14-01974-f008:**
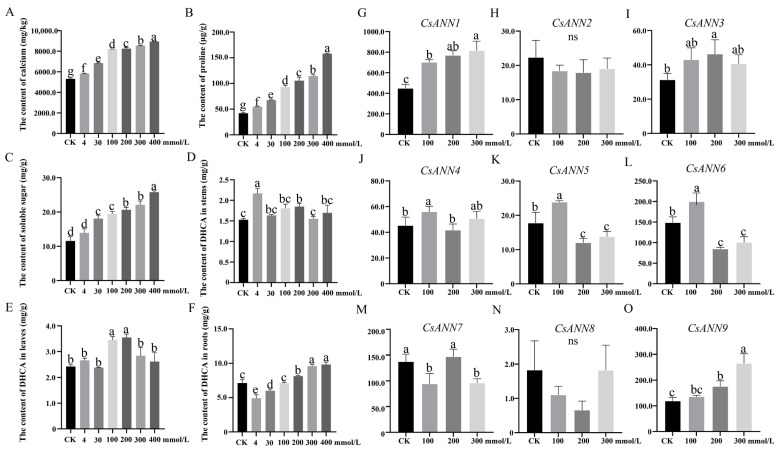
The effects of exogenous CaCl_2_ treatment on *C. saxicola* seedlings. (**A**–**C**): The concentrations of calcium, proline, and soluble sugars in mature leaves of *C. saxicola*, which were exposed to different CaCl_2_ concentrations. (**D**–**F**): The DHCA levels in the stems, mature leaves, and roots of *C. saxicola*, which were treated with different CaCl_2_ concentrations. (**G**–**O**): The relative expression levels of *CsANN*s treated with different concentrations of CaCl_2_. The lowercase letters represent statistically significant differences, and ns represents no statistically significant difference.

**Figure 9 plants-14-01974-f009:**
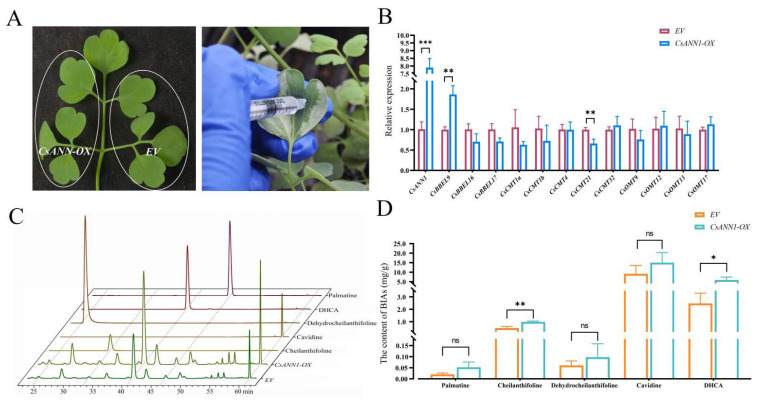
The transient overexpression and functional analysis of *CsANN1*. (**A**) A schematic diagram of the infiltration of *C. saxicola* leaves with either *CsANN*-pCAMBIA1301 or EV; EV: the empty pCAMBIA1301 plasmid vector, serving as a negative control. (**B**) Expression profiles of *CsANN1* and other genes in *C. saxicola* leaves infiltrated with either *CsANN1*-pCAMBIA1301 or EV. (**C**) Changes in BIA concentrations in *C. saxicola* leaves infiltrated with *CsANN1*-pCAMBIA1301 or EV. (**D**) Statistical analysis of BIA levels in *C. saxicola* leaves infiltrated with either *CsANN1*-pCAMBIA1301 or EV. Note: *: *p* < 0.05; **: *p* < 0.01; ***: *p* < 0.001; ns: *p* ≥ 0.05.

**Figure 10 plants-14-01974-f010:**
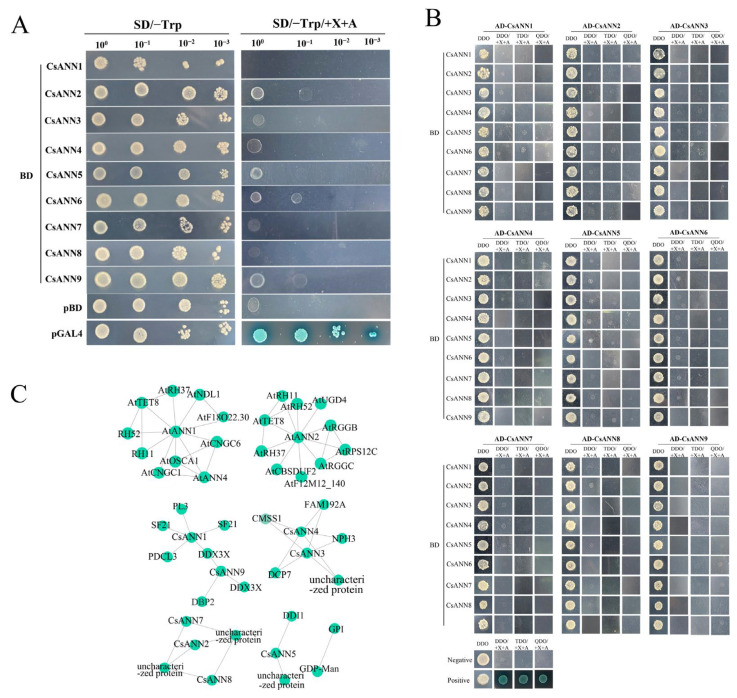
Yeast two hybrid assays and protein interaction networks of the CsANNs. (**A**): Transactivation assay. SD/-Trp: SD medium without Trp; SD/-Trp/+X/+A: SD medium without Trp supplemented with X-α-gal at a concentration of 40 ng/mL; AbA at a concentration of 100 ng/mL. (**B**): Yeast two hybrid assay. DDO: SD medium without Trp and Leu; TDO: SD medium without Trp, Leu, and His; QDO: SD medium without Trp, Leu, His and Ade. (**C**): The protein interaction networks of the CsANNs.

## Data Availability

All relevant data are within the manuscript and its Appendix A.

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
