# Peer review of "Genome-Wide Characterization of the ANN Gene Family in Corydalis saxicola Bunting and the Role of CsANN1 in Dehydrocavidine Biosynthesis"

_plants, 2025, doi:10.3390/plants14131974_

Round 1
Reviewer 1 Report
Comments and Suggestions for Authors
Dear Authors,
Please consider the following points of improvement:
Abstract:
- The term "protein families" is awkward. Annexins are a single family of proteins, not multiple families. Rephrase as “Annexins (ANNs) are a family of Ca²⁺-dependent, phospholipid-binding proteins…”
- “Protection from stresses” is vague. Specify what types of stresses (e.g., biotic, abiotic, oxidative). Biotic or abiotic stresses?
- The importance of studying saxicola is implied but not explicitly justified. Why is it important to understand ANNs in this species? Briefly highlight why this species is a valuable model for studying BIA biosynthesis or stress adaptation.
- No details on what subfamilies these are (e.g., Group I, II, III), and “the phylogenetic analysis” is vague. Clarify the subfamily names or types and mention the reference used for classification.
Introduction:
- Lines 40–42: The first sentence is too long and lists multiple pharmacological properties without proper differentiation or citations per activity. Split into two sentences. Cite specific studies for each pharmacological effect rather than a single general reference.
- The phrase "Yanhuanglian (the Chinese name of C. saxicola)" is ambiguous. Ensure that Yanhuanglian is a widely accepted and specific common name for saxicola. Consider rephrasing as: "locally known as Yanhuanglian" and support with a citation if available.
- The statement that ANNs have peroxidase and ATPase/GTPase activities should be supported with specific studies. Clarify that these enzymatic functions are attributed in specific cases and cite accordingly.
- Lines 64 – 68: Too detailed without integrating relevance to study; overloading the reader with structural info early on. Condense or move structural details (e.g., repeats and motifs) to the Results/Discussion section unless they are directly relevant to the rationale of the study.
- Lines 86 – 94: Objectives are presented but not clearly linked to the knowledge gaps raised earlier. Restructure to first state the knowledge gaps, then the approach, and finally the expected contribution to the field.
Results:
- The subcellular localization predictions are stated as variable, but no quantitative support or confidence values are shown. Include localization prediction confidence scores or percentages from tools like WoLF PSORT or TargetP. Specify which software produced which predictions.
- The phylogenetic tree is based on the NJ method, but there's no mention of bootstrap values, which are crucial for evaluating reliability. Add bootstrap support values to the tree and describe them in the figure legend (e.g., “Numbers at nodes indicate bootstrap values from 1,000 replicates”).
- Many interpretations about function (e.g., peroxidase activity, F-actin binding, GTP binding) are based solely on sequence motifs. Include expression profiles under stress or tissue-specific conditions (RNA-seq or qPCR) to corroborate putative functions.
- The duplication events are described, but their evolutionary significance is not interpreted. Discuss how segmental/tandem duplication may have contributed to functional diversification or subfunctionalization of CsANNs.
- Define all abbreviations on first use.
- You mentioned “Table S1” and “Figure 1A,” but didn’t briefly explain what the reader should observe in those.
Discussion:
- Expand the implications of missing ANN repeats and Ca²⁺ binding sites:
- How exactly might these changes affect signaling or membrane interaction?
- Can these structural deficiencies lead to partial or complete functional loss?
- Provide examples from similar studies in other plants.
- The link between ILAHR motif and peroxidase activity is speculative.
- Has peroxidase activity been biochemically validated in CsANN1, 5, or 9?
- Consider adding: “Further enzymatic assays are needed to confirm this activity in saxicola.”
- You inferred that CsANN1 may regulate CsBBEL9 transcription. This is a strong claim:
- Lacks direct evidence for DNA binding or nuclear localization.
- Better to say: “CsANN1 might be indirectly involved in upregulating CsBBEL9, possibly via interaction with transcriptional regulators.”
- You mentioned cytoplasmic localization and nuclear import under stress:
- Is there localization data under Ca²⁺ stress?
- Suggest verifying with nuclear-cytoplasmic fractionation or live-cell imaging.
- The conclusions regarding CsANN1’s regulatory role are mostly correlative.
- Include functional assays (e.g., RNAi, CRISPR knockout) to strengthen causality.
- Consider transcriptomic profiling of CsANN1-overexpressing plants.
Materials and Methods:
- The number of biological and technical replicates is missing in most experiments.
- Statistical methods used for data analysis are not described.
- Include version numbers of genome assembly and databases used (TAIR, InterPro, SMART).
- Clarify why both HMM and BLAST were used — what's the rationale?
- State how redundancy was resolved in sequence identification.
- State amino acid substitution model used in MEGA (e.g., JTT, Poisson).
- Clarify alignment parameters (e.g., gap penalties, method).
- Mention number of conserved motifs to be identified in MEME (e.g., top 10 motifs).
- Indicate intron/exon structure reference used — genomic DNA or cDNA?
- Define parameters used in collinearity analysis (e.g., minimum match score, max gap).
- Clearly differentiate between tandem, segmental, and whole-genome duplications, if assessed.
- Include whether MCScanX or similar tools were used.
- Provide primer efficiency (ideally between 90–110%).
- Include melt curve analysis or product validation by gel.
- Clarify RNA input amount for cDNA synthesis and qPCR.
- Specify machine model used for qPCR.
- Explain why CsGAPDH8 was selected as the reference gene (any validation?
- Nowhere is it mentioned how data were analyzed statistically. Were t-tests or ANOVAs performed?
- Some control groups are under-described (especially in transgenic and interaction assays).
Convert some passive constructions into active voice to clarify the subject, and there are numerous grammatical errors that need to be fixed.
Author Response
2025.06.20
Dear Editor and Reviewers
Thank you very much for your kindly and professional review concerning our manuscript entitled “Genome-Wide Characterization of the ANN Gene Family in Corydalis saxicola and the Role of CsANN1 in Dehydrocavidine Biosynthesis” (ID: plants-3683213). Those comments are all valuable and very helpful for revising and improving our paper, as well as the important guiding significance to our research. We have studied your comments carefully and have made corrections, which we hope meet with approval. The main corrections in the manuscript and the responses to the comments are as follows:
Reviewer #1:
Abstract:
- The term "protein families" is awkward. Annexins are a single family of proteins, not multiple families. Rephrase as “Annexins (ANNs) are a family of Ca²⁺-dependent, phospholipid-binding proteins…”
Response: Thank you very much for your careful review. We are sorry that we made a mistake here. We have corrected the sentence in the revised manuscript according to your professional suggestion (L20-21). Furthermore, we have also checked the entire manuscript carefully to make sure that no similar mistakes exist.
- “Protection from stresses” is vague. Specify what types of stresses (e.g., biotic, abiotic, oxidative). Biotic or abiotic stresses?
Response: Thank you very much for your professional review. To our knowledge, a number of studies demonstrated that ANNs played crucial roles in the protection of plants from biotic and abiotic stresses (Table A1). We have modified the stresses to biotic and abiotic ones in the revised ‘Abstract’ (L22).
Table A1 The ANNs identified from angiosperm plants.
Species |
Gene numbers |
Developmental and stress responses |
References |
Arabidopsis thaliana |
8 |
Red and far red light; Salt; Drought; High/low temperature |
[1], [2] |
Brassica napus |
23 |
Salt; PEG; Cold; Heat; ABA; ETH; SA; MeJA |
[3] |
Brassica oleracea |
12 |
|
[3] |
Brassica rapa |
13 |
Salt; ABA; SA; MeJA; H2O2 |
[3], [4] |
Carica papaya |
12 |
|
[5] |
Capsicum cannuum |
15 |
SA; PEG; HEAT; COLD; ABA; JA |
[6] |
Cucumis sativus |
11 |
|
[5] |
Glycine max |
23 |
Salt; Drought; Cold; ABA; |
[7] |
Gossypium raimondii |
14 |
Fiber elongation; |
[8] |
Oryza sativa |
10 |
Salt; Drought; High/low temperature; PEG |
[9], [10] |
Phaseolus vulgaris |
13 |
Rhizobial infection |
[11] |
Pisum sativum |
14 |
Rhizobial inoculation |
[12] |
Raphanus sativus |
10 |
Heat; ABA |
[13] |
Solanum lycopersicum |
10 |
Cold; Salt; Drought; Heat; Wound; ABA; GA; SA; IAA; Eth |
[14] |
Solannum tuberosum |
10 |
Drought |
[15] |
Triticum aestivum |
25 |
Salt; PEG; Cold; ABA |
[16] |
Vitis vinifera |
14 |
|
[5] |
Zea mays |
12 |
Heavy metal; JA |
[17] |
[1] Cantero, A.; Barthakur, B.; Bushart, T.J.; Chou, S.; Morgan, R.O.; Fernandez, M.P.; Clark, G.B.; Roux, S.J. Expression profiling of the Arabidopsis annexin gene family during germination, de-etiolation and abiotic stress. Plant Physiol. Biochem. 2006, 44, 13–24.
[2] Clark, G.B.; Sessions, A.; Eastburn, D.J.; Roux, S.J. Differential expression of members of the annexin multigene family in Arabidopsis. Plant Physiol. 2001, 126, 1072–1084.
[3] He, X.; Liao, L.; Xie, S.; Yao, M.; Xie, P.; Liu, W.; Kang, Y.; Huang, L.Y.; Wang, M.; Qian, L.W. Comprehensive analyses of the annexin (ANN) gene family in Brassica rapa, Brassica oleracea and Brassica napus reveals their roles in stress response. Sci. Rep. 2020, 10, 4295.
[4] Yadav, D.; Ahmed, I.; Kirti, P.B. Genome-wide identification and expression profiling of annexins in Brassica rapa and their phylogenetic sequence comparison with B. juncea and A. thaliana annexins. Plant Gene 2015, 4, 109–124.
[5] Jami, S.K.; Clark, G.B.; Ayele, B.T.; Ashe, P.; Kirti, P.B. Genome-wide comparative analysis of annexin superfamily in plants. PLoS One 2012, 7, e47801.
[6] Wu, X.X.; Ren, Y.; Jiang, H.L.; Wang, Y.; Yan, J.X.; Xu, X.Y.; Zhou, F.C.; Ding, H.D. Genome-wide identification and transcriptional expression analysis of annexin genes in Capsicum annuum and characterization of CaAnn9 in salt tolerance. Int. J. Mol. Sci. 2021, 22, 8667.
[7] Feng, Y.M.; Wei, X.K.; Liao, W.X.; Huang, L.H.; Zhang, H.; Liang, S.C.; Peng, H. Molecular analysis of the annexin gene family in soybean. Biol. Plantarum 2013, 57, 655–662.
[8] Tang, W.X.; He, Y.H.; Tu, L.L.; Wang, M.J.; Li, Y.; Ruan, Y.L.; Zhang, X.L. Downregulating annexin gene GhAnn2 inhibits cotton fiber elongation and decreases Ca2+ influx at the cell apex. Plant Mol. Biol. 2014, 85, 613–625.
[9] Jami, S.K.; Clark, G.B.; Ayele, B.T.; Roux, S.J.; Kirti, P.B. Identification and characterization of annexin gene family in rice. Plant Cell Rep. 2012, 31, 813–825.
[10] Singh, A.; Kanwar, P.; Yadav, A.K.; Mishra, M.; Jha, S.K.; Baranwal, V.; Pandey, A.; Kapoor, S.; Tyagi, A.K.; Pandey, G.K. Genome-wide expressional and functional analysis of calcium transport elements during abiotic stress and development in rice. FEBS J. 2014, 281, 894–915.
[11] Carrasco, J.; Ortega, Y.; J´auregui, D.; Ju´arez, M.A.; Arthikala, M.K.; Monroy, E.; Nava, N.; Santana, O.; S´anchez, R.; Quintoa, C. Down-regulation of a Phaseolus vulgaris annexin impairs rhizobial infection and nodulation. Environ. Exp. Bot. 2018, 153, 108–119.
[12] Pavlova, O.A.; Leppyanen, I.V.; Kustova, D.V.; Bovin, A.D.; Dolgikh, E.A. Phylogenetic and structural analysis of annexins in pea (Pisum sativum L.) and their role in legume-rhizobial symbiosis development. Vavilovskii Zhurnal Genet. Selektsii. 2021, 25, 502–513.
[13] Shen, F.; Ying, J.; Xu, L.; Sun, X.; Wang, J.; Wang, Y.; Mei, Y.; Zhu, Y.; Liu, L. Characterization of Annexin gene family and functional analysis of RsANN1a involved in heat tolerance in radish (Raphanus sativus L.). Physiol. Mol. Biol. Plants 2021, 27, 2027–2041.
[14] Lu, Y.; Ouyang, B.; Zhang, J.; Wang, T.; Lu, C.; Han, Q.; Zhao, S.; Ye, Z.; Li, H. Genomic organization, phylogenetic comparision and expression profiles of annexin gene family in tomato (Solanum lycopersicum). Gene 2012, 499, 14–24.
[15] Szalonek, M.; Sierpien, B.; Rymaszewski, W.; Gieczewska, K.; Garstka, M.; Lichocka, M.; Sass, L.; Paul, K.; Vass, I.; Vankova, R.; et al. Potato annexin StANN1 promotes drought tolerance and mitigates light stress in transgenic Solanum tuberosum L. plants. PLoS One 2015, 10, e0132683.
[16] Xu, L.; Tang, Y.; Gao, S.; Hong, L.; Wang, W.W.; Fang, Z.F.; Li, X.Y.; Ma, J.X.; Quan, W.; Li, X.; et al. Comprehensive analyses of the annexin gene family in wheat. BMC Genom. 2016, 17, 415.
[17] Zhou, M.L.; Yang, X.B.; Zhang, Q.; Zhou, M.; Zhao, E.Z.; Tang, Y.X.; Zhu, X.M.; Shao, J.R.; Wu, Y.M. Induction of annexin by heavy metals and jasmonic acid in Zea mays. Funct. Integr. Genomics 2013, 13, 241–251.
- The importance of studying Saxicola is implied but not explicitly justified. Why is it important to understand ANNs in this species? Briefly highlight why this species is a valuable model for studying BIA biosynthesis or stress adaptation.
Response: Thank you for your kindly suggestion. In order to introduce the putative roles of CsANNs play in the BIA-biosynthetic pathway and the adaptation of C. saxicola to calcareous soils in the karst region, we revised the sentences at L22-26 in the revised manuscript.
- No details on what subfamilies these are (e.g., Group I, II, III), and “the phylogenetic analysis” is vague. Clarify the subfamily names or types and mention the reference used for classification.
Response: Thank you for your kind suggestion. We have added the details in the revised manuscript at L27-28. Additionally, we added the subfamily names in Figure 3 in the revised manuscript.
Introduction:
- Lines 40–42: The first sentence is too long and lists multiple pharmacological properties without proper differentiation or citations per activity. Split into two sentences. Cite specific studies for each pharmacological effect rather than a single general reference.
Response: Thank you very much for your professional review. We split the first sentence into two shorter ones based on your suggestion and have altered the references (L43-46) in the revised manuscript.
- The phrase "Yanhuanglian (the Chinese name of saxicola)" is ambiguous. Ensure that Yanhuanglian is a widely accepted and specific common name for saxicola. Consider rephrasing as: "locally known as Yanhuanglian" and support with a citation if available.
Response: Thank you very much for your careful review. In order to ensure the fluency of the sentences, we have added the phrase of "locally known as Yanhuanglian" into the second sentence in the Introduction section (L43-44) in the revised manuscript.
- The statement that ANNs have peroxidase and ATPase/GTPase activities should be supported with specific studies. Clarify that these enzymatic functions are attributed in specific cases and cite accordingly.
Response: Thank you very much for your precise review. We have checked the manuscript thoroughly and deleted the enzymatic functional introduction of ANNs. This was based on following reasons: (1) it seemed that the introduction of enzymatic functions of ANNs here was a little out of place; (2) we have discussed detailed enzymatic functions of ANNs in Section 3.1 in Discussion; (3) the characterization of enzymatic functions of ANNs was not directly relevant to this study.
- Lines 64 – 68: Too detailed without integrating relevance to study; overloading the reader with structural info early on. Condense or move structural details (e.g., repeats and motifs) to the Results/Discussion section unless they are directly relevant to the rationale of the study.
Response: Thank you very much for your suggestion. We have condensed this context at L59-63 in the revised manuscript.
- Lines 86 – 94: Objectives are presented but not clearly linked to the knowledge gaps raised earlier. Restructure to first state the knowledge gaps, then the approach, and finally the expected contribution to the field.
Response: Thank you very much for your professional review. We restructured the sentences at L80-83 of the revised manuscript.
Results
- The subcellular localization predictions are stated as variable, but no quantitative support or confidence values are shown. Include localization prediction confidence scores or percentages from tools like WoLF PSORT or TargetP. Specify which software produced which predictions.
Response: Thank you very much for your professional review. The localization confidence scores of CsANNs as predicted by variable online tools were summarized in Table A2. The results have not been shown in the manuscript due to low importance.
Table A2 The subcellular localizations of CsANNs predicted by variable tools.
Protein ID |
Wolf-psort reliability |
Cell-PLoc 2.0 reliability |
CELLO reliability |
PSORT2 reliability |
Euk-mPLoc 2.0 reliability |
CsANN1 |
cytoplasm 12.570 |
cytoplasm not provided |
cytoplasm 2.646 |
cytoplasm 76.7 |
extracell not provided |
CsANN2 |
cytoplasm 14.051 |
cytoplasm not provided |
nuclear 2.880 |
nuclear 70.6 |
extracell not provided |
CsANN3 |
peroxisome 16.296 |
cytoplasm not provided |
mitochondrial 2.581 |
cytoplasm 76.7 |
extracell not provided |
CsANN4 |
extracell 14.286 |
cytoplasm not provided |
mitochondrial 1.923 |
cytoplasm 65.2 |
extracell not provided |
CsANN5 |
peroxisome 15.972 |
cytoplasm not provided |
cytoplasm 3.463 |
cytoplasm 89.0 |
extracell not provided |
CsANN6 |
extracell 14.296 |
cytoplasm not provided |
mitochondrial 1.923 |
cytoplasm 65.2 |
extracell not provided |
CsANN7 |
cytoplasm 15.142 |
cytoplasm not provided |
cytoplasm 1.973 |
cytoplasm 94.1 |
extracell not provided |
CsANN8 |
nuclear 12.804 |
cytoplasm not provided |
nuclear 2.193 |
nuclear 89.0 |
extracell not provided |
CsANN9 |
cytoskeleton 13.504 |
cytoplasm not provided |
cytoplasm 1.847 |
cytoplasm 70.6 |
extracell not provided |
- The phylogenetic tree is based on the NJ method, but there's no mention of bootstrap values, which are crucial for evaluating reliability. Add bootstrap support values to the tree and describe them in the figure legend (e.g., “Numbers at nodes indicate bootstrap values from 1,000 replicates”).
Response: Thank you very much for your technical suggestion. We have redrawn Figure 2 and revised the legend (L143-145). In addition, we have corrected another mistake of the neglect of CyANN38 in the previous phylogenetic tree. All the modifications are shown in Figure 2. Additionally, the related contents have been also corrected at L130-132 of the revised manuscript.
- Many interpretations about function (e.g., peroxidase activity, F-actin binding, GTP binding) are based solely on sequence motifs. Include expression profiles under stress or tissue-specific conditions (RNA-seq or qPCR) to corroborate putative functions.
Response: Thank you very much for your careful review. Actually, the peroxidase activity, F-actin binding and GTP binding activities of CsANNs were predicted based on the sequence motifs and not the experimental results. We supposed that these functions might not have very close correlations with our studies. As a result, we have not carried out experiments to validate these activities. In addition, the functions of CsANNs predicted based on the RNA-seq or qRT-PCR results under stress or tissue-specific conditions could provide us with some clues that some CsANNs might participate in the BIA biosynthesis and response to high calcium stress. Actually, more solid evidence provided by the experiments summarized in our manuscript have demonstrated the functional authenticity of CsANN1.
- The duplication events are described, but their evolutionary significance is not interpreted. Discuss how segmental/tandem duplication may have contributed to functional diversification or subfunctionalization of CsANNs.
Response: Thank you very much for your professional review. We have to apologize that we made a mistake here. Actually, the collinearity analysis revealed that the presence of CsANN1 and CsANN9 was attributed to whole genome duplication, but not on the segmental duplication. We have corrected this in the revised manuscript. Furthermore, we have briefly interpreted the putative meaning of the different duplicated events that occurred among the CsANNs (L177-184).
- Define all abbreviations on first use.
Response: Thank you very much for your careful review. We have checked the manuscript comprehensively and added the full names of the abbreviations upon their first use (L98, L122, L123, L124, and L128).
- You mentioned “Table S1” and “Figure 1A,” but didn’t briefly explain what the reader should observe in those.
Response: Thank you very much for your careful review. Table S1 summarized the physicochemical analysis results of CsANNs, and we briefly introduced it mainly in Section 2.1. For Figure 1A, we added more details in the revised manuscript at L107-110.
Discussion
- Expand the implications of missing ANN repeats and Ca²⁺ binding sites: How exactly might these changes affect signaling or membrane interaction? Can these structural deficiencies lead to partial or complete functional loss? Provide examples from similar studies in other plants.
Response: Thank you for your kind suggestion. We have read a large number of papers and found that there is relatively little research on the corrlation between ANN repeats’ missing and functional loss of plant ANNs. However, amino acid mutations or certainties in repetitive sequences and Ca2+ binding sites in ANNs altered the functions of the proteins: (1) The phylogenetic analysis of ANNs in the species among eight major phyla by Clark et al revealed that calcium-coordinating residues in repeat 1 are generally more highly conserved than those in repeat 4. He summarized the actual amino acid composition at these sites, and showed marked divergence among many different subfamilies. This ranges from the additional calcium-binding capacity in combination with proximal Cys conservation in early diverging subfamilies ANXD1–D13, to complete loss of predicted calcium-binding capability in at least eight plant subfamiles. This disparity makes it clear that generalizations would be unjustified regarding the mechanism, affinity and capacity of different plant ANNs to bind cell membranes, whereas individual plant ANNs are likely to exhibit very distinct binding properties, subcellular localization kinetics and molecular interactions [1]. (2) Regarding the predicted, fairly well-conserved calcium-binding sites that many plant annexins have in the first or fourth repeat, a study by Lim et al. (1998) found that a substitution of specific acidic residues with an asparagine residue resulted in loss of binding activity only when the change is made in repeat 4, suggesting that this repeat is indeed important in binding calcium [2]. (3) Shin et al found that in contrast to animal ANNs, nucleotide binding and hydrolysis of plant ANNs may depend on a Walker A motif (GXXXXGKT/S) and a GTP-binding motif typical of the GTPase superfamily (DXXG). These sequences have been found in the fourth repeat of AnxGh1; C-terminal deletion and loss of the fourth repeat abolished its GTPase activity [3]. (4) Alignments of the primary sequences of cotton ANN, AnxAt2, and AnxZm33/35 of maize showed that the GTP-binding motifs overlap the Ca2+-binding motif of the fourth endonexin domain. Ca2+ and GTP may therefore compete for binding. Mutagenesis of the Ca2+-binding sites does not impair GTPase activity [4].
[1] Clark, G.B.; Morgan, R.O.; Fernandez, M.P.; Roux, S.J. Evolutionary adaptation of plant annexins has diversified their molecular structures, interactions and functional roles. New Phytol. 2012, 196, 695-712
[2] Lim, E.K.; Roberts M.R.; Bowles D.J. Biochemical characterization of tomato annexin p35, independence of calcium binding and phosphatase activities. J. Biol. Chem. 1998, 273, 34920–34925
[3] Shin, H.S.; Brown, R.M. GTPase activity and biochemical characterization of a recombinant cotton fibre annexin. Plant Physiol. 1999, 119, 925–934.
[4] Calvert, C.M.; Gant, S.J.; Bowles, D.J. Tomato annexins p34 and p35 bind to F-actin and display nucleotide phosphodiesterase activity inhibited by phospholipid binding. Plant Cell 1996, 8, 333-342.
- The link between ILAHR motif and peroxidase activity is speculative. Has peroxidase activity been biochemically validated in CsANN1, 5, or 9? Consider adding: “Further enzymatic assays are needed to confirm this activity in saxicola.”
Response: Thank you very much for your kindly suggestion. Up to now, we have not validated the peroxidase activities in CsANN1, 5, or 9 by biochemical assays. We added “Further enzymatic assays are needed to confirm these activities in C. saxicola.” in L370-371 in the revised manuscript.
- You inferred that CsANN1 may regulate CsBBEL9 transcription. This is a strong claim: Lacks direct evidence for DNA binding or nuclear localization. Better to say: “CsANN1 might be indirectly involved in upregulating CsBBEL9, possibly via interaction with transcriptional regulators.”
Response: Thank you very much for your professional and careful review. We have corrected this in the revised manuscript based on your suggestion (L410-412).
- You mentioned cytoplasmic localization and nuclear import under stress: Is there localization data under Ca²⁺ stress? Suggest verifying with nuclear-cytoplasmic fractionation or live-cell imaging.
Response: Thank you very much for your professional review. References 18 and 19 summarized that some ANNs could move from the cytoplasmic compartment into the nucleus. For example, ANN A2 has a nuclear export signal in its N-terminal domain, but tyrosine phosphorylation might cause conformational changes near this export sequence that allow the protein to enter the nucleus [20]. ANN A5 also seems to enter the nucleus after serum stimulation and tyrosine-kinase signaling [19]. ANN A11 has been detected as a nuclear protein that translocates from the nucleoplasm to the nuclear envelope in cells at prophase [21]. The dynamic localization of ANNs could help RNA transport and export from the nucleus [22,23]. However, to our knowledge, there is no report regarding the dynamic localization study of ANNs under Ca²⁺ stress. As CsANNs could interact with CsMYBs and then this might regulate the biosynthesis of DHCA. We will further verify the localization of these CsANNs in organelles in our future studies.
[18] Volker, G.; Carl, C.; Stephen, M. Annexins: Linking Ca; signalling to membrane dynamics. Nat. Rev. Mol. Cell Biol. 2005, 6, 449–461.
[19] Volker, G.; E, M.S. Annexins: from structure to function. Physiol. Rev. 2002, 82, 331–371.
[20] Eberhard, D. A.; Karns, L. R.; VandenBerg, S. R.; Creutz, C. E. Control of the nuclear-cytoplasmic partitioning of annexin II by a nuclear export signal and by p11 binding. J. Cell Sci. 2001, 114, 3155-3166.
[21] Tomas, A.; Moss, S. E.; Calcium- and cell cycle-dependent association of annexin 11 with the nuclear envelope. J. Biol. Chem. 2003, 278, 20210-20216.
[22] Vedeler, A.; Hollas, H. Annexin II is associated with mRNA which may constitute a distinct subpopulation. Biochem. J.2000, 348, 565-572.
[23] Filipenko, N. R.; Macleod, T. J.; Yoon, C. S.; Waisman, D. M. Annexin A2 is a novel RNA binding protein. J. Biol. Chem. 2004, 279, 8723-8731.
- The conclusions regarding CsANN1’s regulatory role are mostly correlative. Include functional assays (e.g., RNAi, CRISPR knockout) to strengthen causality. Consider transcriptomic profiling of CsANN1-overexpressing plants.
Response: Thank you very much for your professional review. We have added the functional assay context in lines 428-433 in the revised manuscript.
Materials and Methods
- The number of biological and technical replicates is missing in most experiments.
Response: Thank you very much for your careful review. We apologize for our negligence. We added biological and technical replicates in L499-500, L576, L601-602, and 625-626 in the revised manuscript.
- Statistical methods used for data analysis are not described.
Response: Thank you very much for your professional suggestion. We have added this in the revised manuscript at L627-634.
- Include version numbers of genome assembly and databases used (TAIR, InterPro, SMART).
Response: Thank you for your professional review. We have added the information at L502-517 in the revised manuscript.
- Clarify why both HMM and BLAST were used — what's the rationale?
Response: Thank you very much for your professional review. BLAST is a method used to identify genes by comparing the similarities of different amino acid sequences. BLAST can quickly identify the genes with close evolutionary relationships and high similarities in sequence distribution data. However, the sensitivity of BLAST will drop sharply when the gene family members share a relatively low amino acid sequence similarities, thus leading to possible misjudgment of the results. HMM is better at identifying distant homologous genes with the same conserved domains. As a result, by adopting both BLAST and HMM methods we are able to identify the gene members more comprehensively, accurately, and reliably.
- State how redundancy was resolved in sequence identification.
Response: Thank you so much for your technical suggestion. In fact, there was no redundancy in the CsANN sequence identity results when using BLAST and HMM methods. Nevertheless, considering that this step is indispensable in sequence identification, we did not omit it in the manuscript.
- State amino acid substitution model used in MEGA (e.g., JTT, Poisson).
Response: Thank you so much for your technical question. The substitution type in amino acid and the model used was the Poisson model. We have added it in L537-538 in the revised manuscript.
- Clarify alignment parameters (e.g., gap penalties, method).
Response: Thank you for your technical suggestion. The phylogenetic tree of the ANNs was constructed using the neighbor-joining method, with a bootstrap value of 1000 replicates. The analysis was conducted with the amino acid substitution type, pairwise deletion for gap data treatment, and the Poisson model, while other parameters were maintained at their default settings. However, considering that the steps of the evolutionary tree visualized in the previous paper were not clear, we supplemented the details at L535-539 of the revised manuscript.
- Mention number of conserved motifs to be identified in MEME (e.g., top 10 motifs).
Response: Thank you for your careful review. The maximum number of motifs identified by MEME were set up to 10. We added this in L541 in the revised manuscript.
- Indicate intron/exon structure reference used — genomic DNA or cDNA?
Response: Thank you for your review. The gene intron/exon structure of CsANNs were analyzed based on genomic DNA of CsANNs. We have revised it at L542 in the revised manuscript.
- Define parameters used in collinearity analysis (e.g., minimum match score, max gap).
Response: Thank you very much for posing this technical problem. We have added the parameters information of the collinearity analysis at L548-549 in the manuscript. Other parameters of the collinearity analysis are as follows:
Figure A3 The parameters setting of the collinearity analysis
- Clearly differentiate between tandem, segmental, and whole-genome duplications, if assessed.
Response: Thank you for your reminder. We are very sorry that we made a mistake here. In fact, CsANN1 and CsANN9 were attributed to whole genome duplication (WGD and not the segmental duplication. We have corrected this in the revised manuscript (L31, L176-181, and L551). Furthermore, we supplemented the definitions of tandem and whole-genome duplications at L549-552 in the revised manuscript. Finally, after careful verification, we found that there were no segmental involvement in the CsANNs.
- Include whether MCScanX or similar tools were used.
Response: Thank you very much for your kindly question. We have not used these tools.
- Provide primer efficiency (ideally between 90–110%).
Response: Thank you very much for your professional review. We are sorry that we have not plotted the standard curves for qRT-PCR primers of each gene. As a result, we have adopted default efficiencies (100%) for each pairs of the qRT-PCR primers. The relative expression results of CsANN and other gene transcripts were mostly reliable because the upregulation and downregulation trends of these transcripts were similar and this was independent of the amplification efficiencies. We have introduced the procedures of the screening and validation of qRT-PCR primers more clearly for each candidate genes in the revised manuscript. Please see the response for the next question.
- Include melt curve analysis or product validation by gel.
Response: Thank you very much for your professional review. Here, we summarized the procedures of the screening and validation of the qRT-PCR primers for each candidate genes as follows:
- Three candidate pairs of the qRT-PCR primers for each gene were designed using Primer Premier 5 based on the specific regions and excluding the conserved domains. The design standards were as follows: PCR product size: 80~300 bp, primer length: 17~25, Tm: 58~62℃, GC: 40~60%, and the other parameters were set to the default values.
- PCRs were conducted using each pairs of primers and with mixed cDNA of variable tissues of Saxicola as templates. The PCR products were analyzed by electrophoresis. The results are shown in Figure A1. We selected the clear, bright and single bands in the lanes that indicated the primers were mostly satisfactory ones.
Figure A1 Electrophoretic gel analysis of qRT-PCR primer screening of CsANNs
- qRT-PCRs were then conducted using each pairs of primers and with mixed cDNA of variable tissues of Saxicola as templates. The melt curves of the amplification results showing single peaks were selected and the qRT-PCR primers used in this study were screened to be use as the candidate ones (Figure A2).
Figure A2 The melt curves for primer screening of CsANNs in qRT-PCR analysis.
- The analysis of the results of PCR and qRT-PCR above were combined and we then ascertained the primers used in qRT-PCR analysis (Table A3).
Table A3 Information of candidate qRT-PCR primers of CsANNs used in this study,
Gene ID |
Primers No. |
Primer sequences (5'-3') |
Type of primers |
Size of the products |
Yes or No |
CsANN1 |
Primer 1 |
TCTCACAGTTCCTCAATCGGTT |
Forward |
91 |
|
TCAGCTTCTCATTTGTTCCCC |
Reverse |
||||
Primer 2 |
GGTTCTCCGACACGCTATCA |
Forward |
200 |
√ |
|
TCCCCATGCCCAACTAATTC |
Reverse |
||||
Primer 3 |
TCTCAAAGAGCATGTCAGCGA |
Forward |
206 |
|
|
GGAGTGGTCAAGCATTCAACTA |
Reverse |
||||
CsANN2 |
Primer 1 |
GAGATGCTGTTATTGCTAGGAATG |
Forward |
228 |
√ |
TCTGAATGGTGTGACTTGTGTGA |
Reverse |
||||
Primer 2 |
TAATGTTCTGTCTCTATGGATGCTG |
Forward |
107 |
|
|
TGAATGGTGTGACTTGTGTGATG |
Reverse |
||||
Primer 3 |
ATGTTCTGTCTCTATGGATGCTGG |
Forward |
257 |
|
|
GGTGTGACTTGTGTGATGTTGC |
Reverse |
||||
CsANN3 |
Primer 1 |
TCGAAAAGGCAGTGAAAAGTG |
Forward |
288 |
|
CCCAGAAGAGAAAGAAGGAAAGT |
Reverse |
||||
Primer 2 |
TCGAAAAGGCAGTGAAAAGTG |
Forward |
290 |
√ |
|
CTCCCAGAAGAGAAAGAAGGAAA |
Reverse |
||||
Primer 3 |
GCAGAGAATGATGCCAAGGC |
Forward |
88 |
|
|
CGTCCACTAAAAATATGGATGAAGG |
Reverse |
||||
CsANN4 |
Primer 1 |
ATGTCATCTCTGTGTGTTCCTCCT |
Forward |
210 |
|
TTGTTGTTACCACTCAGCTCCTT |
Reverse |
||||
Primer 2 |
ACTCGAAAAGGCAGTGAAAAGTG |
Forward |
297 |
|
|
GCTAGGACCCAGAAGAGAAAGAAG |
Reverse |
||||
Primer 3 |
TGGTTGAGAAGGATGCCAAG |
Forward |
122 |
√ |
|
CAGAACTAACAGCAGCCAAATGT |
Reverse |
||||
CsANN5 |
Primer 1 |
TGTTCAATGCCTCCTCCTCA |
Forward |
159 |
|
CAACCTCCCGGTCAAATCTG |
Reverse |
||||
Primer 2 |
TGTGGTTCCTTCTCCAACCC |
Forward |
273 |
√ |
|
CTTTTTCTTCAAAGCTCCCTTG |
Reverse |
||||
Primer 3 |
TGAGAGACTCAACAAAGCTTTCC |
Forward |
80 |
|
|
TCAGCATTTCTGTGACCCAATA |
Reverse |
||||
CsANN6 |
Primer 1 |
AAGCCAAAAGCCTGGGTG |
Forward |
147 |
√ |
TTGCCGTGTAGTTCCTTGTAAA |
Reverse |
||||
Primer 2 |
ACGCGAATTTGCACGTTTT |
Forward |
289 |
|
|
TCACTTTTGGACCCTCGTATCTA |
Reverse |
||||
Primer 3 |
GGGAAAGAGATGCTCGACTGG |
Forward |
181 |
|
|
ATTGAAGGTGTTAGCGTGATAGG |
Reverse |
||||
CsANN7 |
Primer 1 |
TGGTTACGAGATAAACGCGC |
Forward |
190 |
√ |
CACCCAATAAAGTCTTGGTGATAGA |
Reverse |
||||
Primer 2 |
AGGAGGATCTCATCAAGCGAC |
Forward |
289 |
|
|
GCCAACCAAAGCAACCAAG |
Reverse |
||||
Primer 3 |
TGCATCAACCGATTACAGGG |
Forward |
168 |
|
|
GCCAACCAAAGCAACCAAG |
Reverse |
||||
CsANN8 |
Primer 1 |
TACAACCAAGATCTTCTTCATCTCC |
Forward |
144 |
|
TATCAACAATACCGCCTCCAA |
Reverse |
||||
Primer 2 |
GTGCAATAAGTAACCAGCAAAAG |
Forward |
232 |
√ |
|
AACAATACCGCCTCCAAAAA |
Reverse |
||||
Primer 3 |
GTGCAATAAGTAACCAGCAAAAG |
Forward |
241 |
|
|
AGAGATATCAACAATACCGCCTC |
Reverse |
||||
CsANN9 |
Primer 1 |
TCGGTATGAGGGACCTGAGG |
Forward |
169 |
|
CGAATGTATCGTTGTAGTGGTTTAG |
Reverse |
||||
Primer 2 |
GGTATGAGGGACCTGAGGTGA |
Forward |
158 |
√ |
|
CGTTGTAGTGGTTTAGAGTTGCAA |
Reverse |
||||
Primer 3 |
CGACCATCACTGTTCCTGGA |
Forward |
93 |
|
|
AAGCCCTCATTTGTTCCCC |
Reverse |
In addition, the screening procedures for the other qRT-PCR primers of the genes, including CsBBELs, CsMYBs, CsOMTs, CsCMTs, and CsGAPDH8, were similar to those listed above.
- Clarify RNA input amount for cDNA synthesis and qPCR.
Response: Thank you very much for your careful review. For cDNA synthesis, 1 μg of the extracted total RNA was used to synthesize 20 μL of total single strand cDNA. For qRT-PCR, the cDNA samples were diluted with triple sterilized distilled H2O, and the diluted products (1 µL) were used for qPCR assays (20 µL). We supplemented the amounts of RNA and cDNA at L582 and L593-594 of the revised manuscript.
- Specify machine model used for qPCR.
Response: Thank you very much for your careful review. We have added the information of the qRT-PCR machine model at L595-596 in the revised manuscript.
- Explain why CsGAPDH8 was selected as the reference gene (any validation?)
Response: As an internal reference gene, it is imperative that it concurrently exhibits the following characteristics: (1) a stable expression level with a relatively high expression quantity across various conditions [24]; (2) a lack of tissue specificity, ensuring expression in all tissues and cell types [25]; and (3) the absence of pseudogenes, which prevents the amplification of genomic DNA contamination that could compromise the final experimental results [26]. Due to its possession of these attributes, GAPDH8 is frequently employed as an internal reference gene in many studies. The studies investigating the screening of internal reference genes have identified the GAPDH8 gene as relatively reliable, demonstrating excellent stability [27-29]. Therefore, we also selected GAPDH8 in C. saxicola as the internal reference gene in our studies. However, it should be noted that more candidate internal genes should be strictly screened and validated under various stress conditions using variable evaluation software packages, including geNorm, BestKeeper, NormFinder and RefFinder. We will conduct this analysis in the near future.
[24] Mary, P.; Elaine, P.L.; Jen, N.; Marc, F.; Darlon, V.L.; Jose, O.R.s; Anna, T.; Anand, C.M. Selection of stable internal control genes for quantitative real-time PCR (RT-qPCR) in banana (Musa spp.) under short-term drought stress. Mol. Biol. Rep. 2025, 52, 526.
[25] Peilan, Z.; Shu, C.; Siyu, C.; Yuan, Z.; Yu, L.; Xin, X.; Zhong, L.; Shuang, Z. Selection and Validation of qRT-PCR Internal Reference Genes to Study Flower Color Formation in Camellia impressinervis. Int. J. Mol. Sci. 2023, 9, 1014.
[26] Lu, L.L.; Tang, Y.H.; Xu, H.J.; Qian, Y.; Tao, J.; Zhao, D.Q. Selection and verification of reliable internal reference genes in stem development of herbaceous peony (Paeonia lactiflora Pall.) Physiol. Mol. Biol. Plants 2023, 29, 773-782.
[27] Schmittgen, T.D.; Zakrajsek, B.A. Effect of experimental treatment on housekeeping gene expression: validation by real-time, quantitative RT-PCR. J. Biochem. Biophys. Methods 2000, 46, 1-2.
[28] Kiarash, J.G.; Dayton, W.H.; Amirmahani, F.; Mehdi, M.M.; Zaboli, M.; Nazari, M.; Saeed, M.S.; Jamalvandi, M. Selection and validation of reference genes for normalization of qRT-PCR gene expression in wheat (Triticum durum L.) under drought and salt stresses. J. Genet. 2018, 97, 1433-1444.
[29] Moraes, G.P.; Benitez, L.C.; Amaral, M.N.; Vighi, I.L.; Auler, P.A.; Maia, L.C.; Bianchi, V.J.; Braga, E.J.B. Evaluation of reference genes for RT-qPCR studies in the leaves of rice seedlings under salt. GMR, Genet. Mol. Res. 2015, 14, 2384-2398.
- Nowhere is it mentioned how data were analyzed statistically. Were t-tests or ANOVAs performed?
Response: We sincerely appreciate your professional guidance. We have added “4.11 Statistical Analysis” at L629-631 of the revised manuscript.
- Some control groups are under-described (especially in transgenic and interaction assays)
Response: Thank you very much for your careful review. We are very sorry for our negligence. We have modified and supplemented the information of the control groups as suggested, including details of the transgenic and interaction assays (L612-614 and L622-623).
- Convert some passive constructions into active voice to clarify the subject, and there are numerous grammatical errors that need to be fixed.
Response: We apologize for some confusing text in our previous manuscript and thank you very much for your kindly suggestion. With the help of Dr Dev Sooranna of Imperial College in London, we have polished our manuscript throughly and carefully. We have given our sincere thanks to him in the Acknowledgement Section. We really hope the language of the revised manuscript meets with your approval.
Reviewer #2:
Abstract:
- Figure 1 (A): the meaning of circles should be explained in the figure caption.
Response: Thank you very much for your careful review of our manuscript. We have supplemented the meanings represented by the different circles in the legend of Figure 1 (L112-114).
- Figure 2, the Latin name of species used for phylogenetic tree should be in italic form.
Response: Thank you for your careful review. We are sorry that we made a mistake here. We have corrected this at L145-147 of the revised manuscript. Moreover, we have checked and corrected similar mistakes throughout the revised manuscript.
- Each chart in Figure 3 indicated A, B, C, and D should be included in the caption.
Response: Thank you for your kind suggestions. We have added the A, B, C, and D in the caption of Figure 3 according to your suggestions. Please see L163-164 in the revised manuscript.
- Figure 7, the letter S presents ‘ems’? I think it should be ‘Stems’.
Response: Thank you for your careful review. We have corrected this mistake at L234 in the revised manuscript.
- In section 2.7, some indicators including calcium content, proline content, soluble sugar content, DHCA content were analyzed to test the effects of CaCl2 treatment, but how to determine them? The measurement methods were not found in Materials and Methods.
Response: Thank you for your reminder. We have supplemented the determination methods for the levels of calcium, proline, and soluble sugars at L572-576 in the revised “Materials and Methods”. In addition, the DHCA measurement method for DHCA and other BIA were also been described in Section 4.6 at L560-571.
- The expression profiles of CsANNs were determined according to the transcriptomic data. However, there's no explanation of how the transcriptomic data was obtained.
Response: Thank you very much for your careful review. The acquisition procedure of the transcriptomic data was as follows: Total RNA was extracted from C. saxicoia roots using an ethanol precipitation protocol and CTAB-pBIOZOL reagent. RNA quantity and purity were assessed by using a Nanodrop micro-spectrophotometer and an Agilent 2100 Bioanalyzer (Agilent), respectively. RNA was fragmented and reverse-transcribed using random N6 primers to synthesize cDNA (BGI Geno-mics, Shenzhen, China). The ends of double-stranded DNA were incubated with dUTP to form flat ends. The 5’-UTR was phosphorylated, and the 3’-UTR formed cohesive ends with A-tailing and they were connected to the 3’-UTR with T-tailed ends. A cDNA library was constructed using PCR amplification for Illumina RNA sequencing. The raw data were filtered by using SOAPnuke v1.5.6 to remove the reads containing sequencing adapters, those with low-quality bases (base quality ≤15) ratio is >20 % and the reads with unknown base ratios of >5 %.
As the sequencing and data acquisition of the transcriptome are not the main research methods involved in this article, we did not include these in the revised manuscript.
- In Figure 9, the EV representing empty vector should be stated in the caption.
Response: Thank you very much for your professional review. We added the information of EV in the legend of Figure 9 based on your suggestion at L290-291.
- Y2H assay revealed that neither CsANNs had no auto-activation activities or could form homodiers or heterodimers. In Discussion section, the author indicated that CsANNs could interact with CsMYBs, but why the data was not shown in the manuscript? If these data were incorporated into the Y2H assay, it will be more convincing for the function of CsANNs involved in the biosynthesis of DHCA.
Response: Thank you for your friendly question. According to transcriptome data and correlation analysis results, we identified two candidate MYBs, CsMYB10 and CsMYB24, that might participate in the biosynthesis of DHCA. Y2H assays demonstrated that CsANN9 could interact with CsMYB24 in yeast cells (Figure A3). To confirm the correlation of CsANNs and CsMYBs in plants, transient overexpression assays of CsANNs and/or CsMYBs were conducted in C. saxicola leaves. Overexpression of CsMYBs could decrease the content of DHCA in C. saxicola leaves, whether the CsANNs were overexpressed or not (Figure A4). In addition, overexpression of CsANN1 could alleviate the negative effects of CsMYBs in DHCA biosynthesis, while co-overexpression of CsANN9 and CsMYBs displayed the complicated effects in DHCA biosynthesis towards solely overexpression CsMYB10 or CsMYB24 (Figure A4). We speculated that CsANNs and CsMYBs could regulate the biosynthesis of DHCA in C. saxicola leaves. However, more robust evidence is needed in order to uncover their relationships at the molecular level during this regulation processes. We will expand and extrapolate this work in our future endeavors.
Figure A3 Yeast two hybrid assay and protein interaction networks of the CsANNs and CsMYBs. SD/-Trp: SD medium without Trp, SD/-Trp/+X/+A: SD medium without Trp supplemented with X-α-gal at a concentration of 40 ng/mL, AbA at a concentration of 100 ng/mL. DDO: SD medium without Trp and Leu; TDO: SD medium without Trp, Leu, and His; QDO: SD medium without Trp, Leu, His and Ade.
Figure 4A The transient overexpression and functional analysis of CsANNs and/or CsMYBs. The content of BIAs were determined by HPLC.
- In 4.2, The access number of genome of C. saxicaola should be stated.
Response: Thank you for your friendly question. The genome data of C. saxicaola was submitted to the NCBI website (https://www.ncbi.nlm.nih.gov/datasets/genome/GCA_047716255.1/). We have added the address of the website in L515-517 of the revised manuscript.
- The web address of online websites including Wolf-psort, Cell-PLox, PSORTS2, Euk-mPLoc should be added.
Response: Thank you very much for your carefully review. We have added the websites in L526-530 of the revised manuscript.
- Please verify the format of the references. Are all initial letters of words in the title capitalized?
Response: Thank you very much for your careful inspection. We are very sorry that we made a mistake here. We have corrected the references in accordance with the standard format.
Reviewer #3:
Abstract:
- Pay attention to the clarity of language: Line23: BIA-riched → BIA-rich; The word ‘show’ has been used many times:
Response: Thank you for your suggestion. We have made the revisions as per your suggestion (L23). Additionally, we have checked the entire text to ensure there are no similar errors. We are sorry that some errors and confusing text in our previous manuscript. We have polished our manuscript throughly and carefully. We really hope the language of the revised manuscript meets with your approval.
- Line 29:The CsANN transcripts showed variable organ-specific and Ca2+-responsive expression patterns. Modified version:The CsANN transcripts displayed organ-specific and Ca2+-responsive expression patterns across various tissues.
Response: Thank you very much for your careful review. We have changed the sentence at L32 in the revised manuscript according to your suggestion.
Figure:
- In Figure 7, the last label on the x-axis should be 'F' instead of 'FP’.
Response: Thank you for your professional advice. We are very sorry for our negligence. We have corrected Figure 7 based on your comment.
- Line 227, ‘S’ is supposed to mean ‘stems’, not ‘ems’
Response: Thank you for your careful review. We have corrected this mistake at L234 in the revised manuscript.
Discussion:
- What is the difference of CsANN between Arabidopsis and saxicola?
Response: Thank you for your kindly suggestion. (1) The numbers are different, with Arabidopsis possessing eight ANNs compared to nine in C. saxicola. This kind of difference, however, underscores the notion that within a single plant species, ANNs may be a diverse polygenic protein family. (2) In comparison to AtANNs, the predicted molecular masses of proteins encoded by CsANNs exhibit greater variability, ranging approximately from 35.48 kDa to 38.67 kDa. Conversely, the predicted molecular mass of all proteins encoded by AtANNs is consistently around 36 kDa. (3) Compared with AtANN1, there are short insertions found within the structural repeats of these ANNs in of two plant species. For example, AtANN4 has an 11-amino acid insertion in its first repeat, and AtANN3 has a five-amino acid insertion in its second repeat. CsANN2 has a 2-amino acid insertion in its third repeat, and CsANN5 has a 3-amino acid insertion in its second repeat. (4) Unlike CsANNs, the Ca2+ binding sites of AtANNs are more conserved, especially in fourth repeat. Examination of the AtANNs for the presence of type II Ca2+ binding sites revealed that AtANN1, AtANN2, AtANN6, and AtANN7 have fairly well conserved Ca2+ binding sites in their first and fourth repeat. (5) The primary functional binding sites exhibit differences. Specifically, the DXXG site, which has the potential to bind to GTP, is present in six AtANNs, whereas it is found in only three CsANNs. (6) AtANNs and CsANNs have different genetic structural characteristics. Specifically, AtANN3, AtANN4, AtANN5 and AtANN8 have similar genetic structures, and each gene contains six exons. However, the structures of CsANN1 and CsANN9 are quite similar, with each gene containing five exons. CsANN4 and CsANN6 are also quite similar, with each gene containing six exons. Some AtANNs have lost some introns. AtANN1 has lost the final three introns, AtANN2 is missing a single final intron, and the tandem pair members, AtANN6 and AtANN7, have lost their final two introns. Unlike AtANNs, CsANN2, 6, and 7 have 5 introns, CsANN9 has 6, and the rest have 4. This indicates that CsANN3 is a recently evolved gene and may have new functions. In contrast, the genetic structure of AtANNs seems to be more conformed, indicating its earlier origin [1,2].
[1] Clark, G.B.; Sessions, A.; Eastburn, D.J.; Roux, S.J. Differential expression of members of the annexin multigene family in Arabidopsis. Plant Physiol. 2001, 126, 1072-1084.
[2] Cantero, A.; Barthakur, S.; Bushart, T.J.; Chou, S.; Morgan, R.O.; Fernandez, M.P.; Clark, G.B.; Roux, S.J. Expression profiling of the Arabidopsis annexin gene family during germination, de-etiolation and abiotic. Plant Physiol. Biochem. 2006, 44, 13-24.
- Line 329: but the informations about ANN gene families in medicinal plants were scarce. The word ‘information’ is an uncountable noun.
Response: Thank you very much for your suggestion. We are sorry that we made a mistake here. We have corrected this (L339) and checked and corrected similar errors in the revised text.
- Line 351 & 352: AnnAt1 → AtAnn 1 AnnBr1 → BrAnn 1
Response: We really appreciate your careful examination. We have corrected these two mistakes at L361 and L362 in the revised manuscript.
- Line 402: You have mentioned ‘subcellular localization analyses of CsANN1 and CsANN9 suggested a more likely presence in the cytoplasm’, but no picture has been shown in your assay.
Response: Thank you very much for your technical suggestion. The subcellular localization analysis of CsANNs were displayed in Figure 1A and they are illustrated in L107-110 in the revised manuscript. Accordingly, we have made revisions and cited Figure 1A in L415-416.
Conclusions:
- Line 577: with closely related members having similar structural characteristics. Modified version for clarity: with closely related members sharing structural similarities.
Response: Thank you very much for your professional suggestion. We have made corrections in L638-639 of the revised article based on your suggestion.
- Line 577: In addition, gene structures, conserved motifs, chromosomal distributions, collinearities, protein interactions, and cis-acting elements of CsANN genes or CsANN proteins were also analyzed. Modified version: In addition, various genomic and proteomic features—including gene structures, conserved motifs, chromosomal distributions, collinearity, protein interactions, and cis-acting elements—were systematically characterized.
Response: Thank you very much for your professional comments. We have made corrections in L639-641 of the revised manuscript based on your suggestion.
Editor:
Thank you very much for your technical comments. We have made revisions according to you and the reviewers’ kindly suggestions.
- Ensure all references are relevant to the content of the manuscript.
Response: Thank you very much for your professional reminder. We conducted a thorough review of the references cited in the manuscript and found several inaccuracies. Consequently, we removed extraneous references, incorporated additional new pertinent citations, reordered the references accordingly, and revised the formatting to ensure accurate citation of each source.
- Highlight any revisions to the manuscript, so editors and reviewers can see any changes made.
Response: Thank you for your professional reminder. To enable the Editor and Reviewers to clearly identify our revisions, we utilized the revision mode to modify the manuscript.
- Provide a cover letter to respond to the reviewers’ comments and explain, point by point, the details of the manuscript revisions.
Response: Thank you very much for your professional reminder. We have written a cover letter to explain the details of how we revised the manuscript in accordance with Editor and Reviewers' comments.
- If the reviewer(s) recommended references, critically analyze them to ensure that their inclusion would enhance your manuscript. If you believe these references are unnecessary, you should not include them.
Response: Thank you very much for your kindly comments. The reviewers did not recommend any additional references.
- If you found it impossible to address certain comments in the review reports, include an explanation in your appeal.
Response: Thank you very much for your kind help. All the comments for this manuscript are reasonable and we have tried our best to correct our manuscript according to the Reviewers’ suggestions. We hope the revised manuscript meets with your approval.
Best regards,
Han Liu and other co-authors

Reviewer 2 Report
Comments and Suggestions for Authors
In this study, nine CsANNs were firstly identified and characterized. Secondly, authors demonstrated that it was CsANN1, but not CsANN9, could increase the content of DHCA in C. saxicola leaves. The results were vital for understanding the biological functions of CsANNs especially CsANN1. However, there are some comments authors should be concerned.
- Figure 1 (A): the meaning of circles should be explained in the figure caption.
- Figure 2, the latin name of species used for phylogenetic tree should be in italic form.
- Each chart in Figure 3 indicated A, B, C, and D should be included in the caption.
- Figure 7, the letter S presents ‘ems’? I think it should be ‘Stems’.
- In section 2.7, some indicators including calcium content, proline content, soluble sugar content, DHCA content were analyzed to test the effects of CaCl2 treatment, but how to determine them? The measurement methods were not found in Materials and Methods.
- The expression profiles of CsANNs were determined according to the transcriptomic data. However, there's no explanation of how the transcriptomic data was obtained.
- In Figure 9, the EV representing empty vector should be stated in the caption.
- Y2H assay revealed that neither CsANNs had no auto-activation activities or could form homodiers or heterodimers. In Discussion section, the author indicated that CsANNSs could interact with CsMYBs, but why the data was not shown in the manuscript? If these data were incorporated into the Y2H assay, it will be more convincing for the function of CsANNs involved in the biosynthesis of DHCA.
- In 4.2, The access number of genome of C. saxicaola should be stated.
- The web address of online websites including Wolf-psort, Cell-PLox, PSORTS2, Euk-mPLoc should be added.
- Please verify the format of the references. Are all initial letters of words in the title capitalized?
Author Response

(The authors gave the same response as above.)

Reviewer 3 Report
Comments and Suggestions for Authors
This MS identifies C. saxicola CsANN and demonstrates that CsANN1 could positively regulate the accumulation of BIA compounds.
Abstract:
Pay attention to the clarity of language:
Line23: BIA-riched → BIA-rich
The word ‘show’ has been used many times:
Line 29:The CsANN transcripts showed variable organ-specific and Ca2+-responsive expression patterns.
Modified version:The CsANN transcripts displayed organ-specific and Ca2+-responsive expression patterns across various tissues.
Figure:
In Figure 7, the last label on the x-axis should be 'F' instead of 'FP'.
Line 227, ‘S’ is supposed to mean ‘stems’, not ‘ems’.
Discussion:
What is the difference of CsANN between Arabidopsis and C. saxicola?
Line 329: but the informations about ANN gene families in medicinal plants were scarce.
The word ‘information’ is an uncountable noun.
Line 351 & 352: AnnAt1 → AtAnn 1 AnnBr1 → BrAnn 1
Line 402: You have mentioned ‘subcellular localization analyses of CsANN1 and CsANN9 suggested a more likely presence in the cytoplasm’, but no picture has been shown in your assay.
Conclusions:
1.Line 577: with closely related members having similar structural characteristics.
Modified version for clarity: with closely related members sharing structural similarities.
Line 577: In addition, gene structures, conserved motifs, chromosomal distributions, collinearities, protein interactions, and cis-acting elements of CsANN genes or CsANN proteins were also analyzed.
Modified version: In addition, various genomic and proteomic features—including gene structures, conserved motifs, chromosomal distributions, collinearity, protein interactions, and cis-acting elements—were systematically characterized.
Author Response

(The authors gave the same response as above.)

Round 2
Reviewer 1 Report
Comments and Suggestions for Authors
Dear Authors,
I reviewed the revised version of your manuscript. I found significant improvements in the content and quality. However, I would like you to do the following minor revision:
- Please improve the introduction based on my previous comments.
- Figure 3 - 6: The figure's caption should provide a comprehensive explanation of the figure.
- Figure 7: Please include all sub-figures in the caption.
Author Response
2025.06.23
Dear Editor and Reviewers,
Thank you again for your kindly and professional review concerning our manuscript entitled “Genome-Wide Characterization of the ANN Gene Family in Corydalis saxicola and the Role of CsANN1 in Dehydrocavidine Biosynthesis” (ID: plants-3683213). Those comments are all valuable and very helpful for revising and improving our paper, as well as the important guiding significance to our research. We have studied your comments carefully and have made corrections, which we hope meet with approval. The main corrections in the manuscript and the responses to the comments are as follows:
Reviewer #1:
- Please improve the introduction based on my previous comments.
Response: Thank you very much for your careful review. We have restructured the last paragraph of the Introduction section according to your suggestion. Please see L69-79 in the revised manuscript.
- Figure 3 - 6: The figure's caption should provide a comprehensive explanation of the figure.
Response: Thank you very much for your suggestion. We have added comprehensive explanations of the figures in the revised manuscript.
- Figure 7: Please include all sub-figures in the caption.
Response: Thank you for your professional suggestion. Actually, the letter numbers are unnecessary in Figure 7. We have deleted the letter numbers and updated the annotations in Figure 7.
Best regards,
Han Liu and other co-authors

Reviewer 2 Report
Comments and Suggestions for Authors
The authors studied the effects of CaCl2 treatment on C. Saxicola seedlings. However, either how this experiment was conducted or how the transcriptomic data was obtained was not explained in Materials and method section.
Author Response
2025.06.23
Dear Editor and Reviewers,
Thank you again for your kindly and professional review concerning our manuscript entitled “Genome-Wide Characterization of the ANN Gene Family in Corydalis saxicola and the Role of CsANN1 in Dehydrocavidine Biosynthesis” (ID: plants-3683213). Those comments are all valuable and very helpful for revising and improving our paper, as well as the important guiding significance to our research. We have studied your comments carefully and have made corrections, which we hope meet with approval. The main corrections in the manuscript and the responses to the comments are as follows:
Reviewer #2:
- The authors studied the effects of CaCl2treatment on Saxicola seedlings. However, either how this experiment was conducted or how the transcriptomic data was obtained was not explained in Materials and method section.
Response: Thank you for your review. For the CaCl2 treatment experiment, we already described it at L392-398 in Section 4.1. For the transcriptomic analysis, we have described it at L399-408 in the revised manuscript. Three additional references cited here were also added in the References Section.
Best regards,
Han Liu and other co-authors

Round 3
Reviewer 2 Report
Comments and Suggestions for Authors
It can be accepted.